# Rho GTPase signaling and mDia facilitate endocytosis via presynaptic actin

Kristine Oevel[1], Svea Hohensee[1], Atul Kumar[2], Irving Rosas-Brugada[1†], Francesca Bartolini[2], Tolga Soykan[1]*, Volker Haucke[1,3,4]*

[1]Leibniz-Forschungsinstitut für Molekulare Pharmakologie (FMP), Berlin, Germany; [2]Department of Pathology and Cell Biology, Columbia University Medical Center, New York City, United States; [3]Faculty of Biology, Chemistry, Pharmacy, Freie Universität Berlin, Berlin, Germany; [4]NeuroCure Cluster of Excellence, Charité Universitätsmedizin Berlin, Berlin, Germany

*For correspondence:
soykan@fmp-berlin.de (TS);
haucke@fmp-berlin.de (VH)

†Deceased

**Abstract** Neurotransmission at synapses is mediated by the fusion and subsequent endocytosis of synaptic vesicle membranes. Actin has been suggested to be required for presynaptic endocytosis but the mechanisms that control actin polymerization and its mode of action within presynaptic nerve terminals remain poorly understood. We combine optical recordings of presynaptic membrane dynamics and ultrastructural analysis with genetic and pharmacological manipulations to demonstrate that presynaptic endocytosis is controlled by actin regulatory diaphanous-related formins mDia1/3 and Rho family GTPase signaling in mouse hippocampal neurons. We show that impaired presynaptic actin assembly in the near absence of mDia1/3 and reduced RhoA activity is partly compensated by hyperactivation of Rac1. Inhibition of Rac1 signaling further aggravates impaired presynaptic endocytosis elicited by loss of mDia1/3. Our data suggest that interdependent mDia1/3-Rho and Rac1 signaling pathways cooperatively act to facilitate synaptic vesicle endocytosis by controlling presynaptic F-actin.

## eLife assessment

This manuscript provides **convincing** evidence for the involvement of membrane actin, and its regulatory proteins, mDia1/3, RhoA, and Rac1 in the mechanism of synaptic vesicle re-uptake (endocytosis). These **important** data fill a gap in the understanding of how the regulation of actin dynamics and endocytosis are linked. The manuscript will be of interest to all scientists working on cellular trafficking and membrane remodeling.

## Introduction

Synaptic transmission relies on the release of neurotransmitters by calcium-triggered exocytic fusion of synaptic vesicles (SVs) at specialized active zone release sites within presynaptic nerve terminals. Following fusion, compensatory endocytosis retrieves SV proteins and lipids from the presynaptic plasma membrane and SVs are reformed (*Chanaday et al., 2019*; *Gan and Watanabe, 2018*; *Kononenko and Haucke, 2015*; *Saheki and De Camilli, 2012*; *Soykan et al., 2016*), e.g., by clathrin-mediated vesicle budding (*Kononenko et al., 2014*; *Watanabe et al., 2014*) (but see *Wu et al., 2014* for evidence in favor of clathrin-independent SV reformation). Endocytosis relies on mechanical forces to enable membrane deformation and eventually fission (*Engqvist-Goldstein and Drubin, 2003*). In many biological systems, actin and actin-associated myosin motors appear to provide such mechanical force to facilitate vesicle formation (*Anes et al., 2003*; *Marston et al., 2003*). For example, the function of actin for endocytosis is well-established in yeast, whereas differential requirements

for actin in distinct forms of endocytosis have been described in various types of mammalian cells including neurons (*Boulant et al., 2011*; *Merrifield et al., 2005*; *Saffarian et al., 2009*; *Saheki and De Camilli, 2012*). Pharmacological inhibition of actin assembly by latrunculin has been shown to cause the accumulation of endocytic intermediates at lamprey giant synapses (*Shupliakov et al., 2002*) and interfere with ultrafast endocytosis in response to single optical action potentials (APs) at hippocampal synapses (*Watanabe et al., 2013*; *Watanabe et al., 2014*; *Wu and Chan, 2022*) and with fast endocytosis at cerebellar mossy fiber boutons (*Delvendahl et al., 2016*). The same treatment does not hamper endocytosis of SV proteins at hippocampal synapses stimulated with trains of APs (*Sankaranarayanan et al., 2003*; *Soykan et al., 2017*). Interference with actin function by conditional knockout (KO) of *Actb* or *Actg1* genes, encoding β- or γ-actin, respectively, has suggested a crucial role for actin in all kinetically distinguishable forms of endocytosis (*Wu et al., 2016*) at hippocampal synapses and the calyx of Held, a fast giant synapse in the auditory brain stem (*Borst and Soria van Hoeve, 2012*). However, in the same preparations, actin loss also affected SV exocytosis (*Wu et al., 2016*), suggesting a more general requirement of actin for presynaptic function. Consistently, actin has been shown to surround clusters of reserve pool SVs in lamprey (*Bloom et al., 2003*), to facilitate the replenishment of fast-releasing vesicles in various models (*Sakaba et al., 2013*; *Sakaba and Neher, 2003*), to be involved in Tau pathology (*Zhou et al., 2017*), to induce spine growth and plasticity, (*Cingolani and Goda, 2008*) and to steer neuronal migration (*Shinohara et al., 2012*) among various other roles.

We have previously demonstrated that pharmacological interference with the function of formins, a group of Rho family-associated proteins that nucleate linear filamentous (F) actin, kinetically delays SV endocytosis at hippocampal boutons and blocks compensatory endocytosis at the calyx of Held in pre-hearing rats (*Soykan et al., 2017*). Capacitance measurements at the calyx of Held in post-hearing rodents suggest a less stringent requirement for formin-mediated actin assembly for endocytosis (*Hori et al., 2022*). The latter observation may reflect the operation of compensatory pathways for actin assembly, e.g., via Rac1 or Cdc42 Rho-family GTPases that promote branched actin networks (*Eisenmann et al., 2005*; *Goode and Eck, 2007*; *Hodge and Ridley, 2016*).

In contrast to the vast body of literature regarding the role of actin and actin-associated proteins at the postsynapse (*Cingolani and Goda, 2008*; *Colgan and Yasuda, 2014*) and references therein, most notably the actin-rich mesh of the postsynaptic density of glutamatergic synapses, comparably little is known about the signaling pathways e.g., via guanine nucleotide exchange factors and GTPase activating proteins for Rho family small GTPases (*Müller et al., 2020*) that mediate actin assembly at the presynapse. Recent work has suggested that presynaptic Rac1 negatively regulates synaptic strength and release probability by altering SV priming and replenishment at central synapses (*O'Neil et al., 2021*) including the calyx of Held (*Keine et al., 2022*). These data suggest a negative regulatory role for Rac1 in SV exocytosis. The abundance of postsynaptic actin (*Chen et al., 2020*; *Cingolani and Goda, 2008*; *Colgan and Yasuda, 2014*) has hampered the analysis of the nanoscale localization of actin at presynaptic nerve terminals and of the actin-regulatory proteins that control its dynamics. Loss of dynamin isoforms Dynamin1 and Dynamin3, the main enzymes for endocytic membrane fission (*Ferguson et al., 2007*; *Imoto et al., 2022*; *Raimondi et al., 2011*), has been shown to cause the accumulation of F-actin at and around stalled endocytic intermediates in non-neuronal cells (*Ferguson et al., 2009*) but this phenotype seems less overt at hippocampal synapses (*Raimondi et al., 2011*). Hence, actin in addition to a possible local role at endocytic invaginations might serve a more general function in endocytosis at synapses, for example by modulating plasma membrane tension or contractility, by providing a restrictive membrane scaffold that promotes membrane fission by keeping endocytic invaginations under longitudinal tension (*Boulant et al., 2011*; *Roux et al., 2006*; *Wu and Chan, 2022*), or via formation of an F-actin ring around the active zone that mechanically couples exocytic membrane compression to endocytic pit formation (*Ogunmowo et al., 2023*).

Here, we combine optical recordings of SV exo-/endocytosis and ultrastructural analysis with genetic and pharmacological manipulations to show that interdependent mDia1/3-Rho and Rac1 signaling pathways cooperate to facilitate SV endocytosis by controlling presynaptic F-actin.

## Results

### Actin dynamics and actin nucleating mDia1/3 proteins facilitate presynaptic endocytosis and SV recycling

Previous studies using pharmacological inhibitors of actin polymerization and depolymerization have yielded inconclusive, often conflicting data regarding the function of actin in SV endocytosis and recycling in different models (*Bleckert et al., 2012*; *Del Signore et al., 2021*; *Piriya Ananda Babu et al., 2020*; *Sankaranarayanan et al., 2003*; *Shupliakov et al., 2002*; *Wu and Chan, 2022*). In contrast, ablation of *Actb* or *Actg1*in mouse neurons suggests that actin is required for all forms of endocytosis at several types of synapses (*Wu et al., 2016*). Prompted by these findings we revisited the role of actin in presynaptic endocytosis by analyzing the effects of impaired F-actin dynamics in the combined presence of the G-actin sequestering drug latrunculin A, the F-actin stabilizer jasplakinolide, and Y-27632, an inhibitor of ROCK kinase signaling. This cocktail was shown to preserve the existing cytoskeleton architecture while blocking actin assembly, disassembly, and rearrangement (*Peng et al., 2011*). We optically recorded the stimulation-induced exo-endocytosis of the SV protein Synaptophysin (Syph) fused to a pH-sensitive super-ecliptic green fluorescent protein (pHluorin) (*Miesenböck et al., 1998*) that is de-quenched during exocytosis and undergoes re-quenching as SVs are internalized and re-acidified in hippocampal neurons at physiological temperature. Under these conditions (i.e. trains of APs, 37 °C), SV endocytosis occurs on a timescale of >10 s, e.g., a timescale that is slower than re-quenching of pHluorin due to reacidification (*López-Hernández et al., 2022*; *Soykan et al., 2017*). Hence, the decay of pHluorin signals can serve as a measure of the time course of SV endocytosis. Perturbation of actin dynamics in the combined presence of latrunculin A, jasplakinolide, and Y-27632 significantly slowed down the endocytic retrieval of exogenously expressed Syph-pHluorin (*Figure 1A and B*). Application of Y-27632 alone or a combination of latrunculin A and jasplakinolide had no effect on SV endocytosis kinetics, whereas combined use of Y-27632 together with jasplakinolide displayed a mild inhibitory effect (*Figure 1—figure supplement 1C–F*). Impaired actin dynamics in the presence of latrunculin A, jasplakinolide, and Y-27632 did not impact the apparent level of SV exocytosis indicated by the maximal amplitude of Syph-pHluorin fluorescence (*Figure 1—figure supplement 1A*). As optical imaging of pHluorin reporters may lead to artifacts, we analyzed the internalization kinetics of the endogenous vesicular γ-aminobutyric acid transporter (vGAT) using antibodies directed against its luminal domain coupled to the pH-sensitive fluorophore CypHer5E (*Hua et al., 2011*; *López-Hernández et al., 2022*). The cyanine-based dye CypHer5E is quenched at neutral pH but exhibits bright fluorescence when present in the acidic lumen of SVs and, thus can serve as a tracer for the exo-endocytic cycling of endogenous SV proteins. Perturbation of actin dynamics significantly delayed the endocytic retrieval of endogenous vGAT in response to train stimulation with 200 APs at physiological temperature (*Figure 1C and D*), consistent with the results from exogenously expressed Syph-pHluorin. vGAT exocytosis proceeded unperturbed (*Figure 1—figure supplement 1B*). Hence, a dynamic actin cytoskeleton facilitates presynaptic endocytosis in hippocampal neurons.

Previous studies have suggested that formins, in particular mDia1, promote SV endocytosis (*Soykan et al., 2017*), likely by inducing F-actin assembly (*Ganguly et al., 2015*). We confirmed that shRNA-mediated depletion of *Diaph1*, encoding mDia1 (shmDia1), from hippocampal neurons led to significantly delayed Syph-pHluorin endocytosis and re-acidification in response to stimulation with 200 APs. Re-expression of shRNA-resistant wild-type mDia1 fully restored normal endocytosis kinetics (*Figure 1E and F*). Impaired SV endocytosis in lentivirus-mediated mDia1-depleted neurons (*Figure 1—figure supplement 1H*) was also observed, if SV exo-/endocytosis was triggered by stimulation with 40 APs or 80 APs and probed with vesicular glutamate transporter 1 (vGLUT1)-pHluorin (*Figure 1G and H* and *Figure 1—figure supplement 1I and K*). Exocytic fusion of vGLUT1-pHluorin-containing SVs was unaffected (*Figure 1—figure supplement 1G*). Co-depletion of the closely related mDia3 isoform tended to further aggravate this phenotype (*Figure 1G and H* and *Figure 1—figure supplement 1I and K*) without impacting the apparent number of fused SVs (*Figure 1—figure supplement 1J and L*) or vGAT exocytosis (*Figure 1—figure supplement 1M*). Furthermore, mDia1/3 depletion significantly delayed the endocytic retrieval of endogenous vGAT (*Figure 1I and J*). Conversely, the application of IMM-01, a small molecule activator of mDia-related formins that acts by relieving autoinhibition (*Lash et al., 2013*; *Figure 1—figure supplement 1N*), led to a moderate but significant acceleration of SV endocytosis monitored by vGLUT1-pHluorin (*Figure 1K and L*) but no change

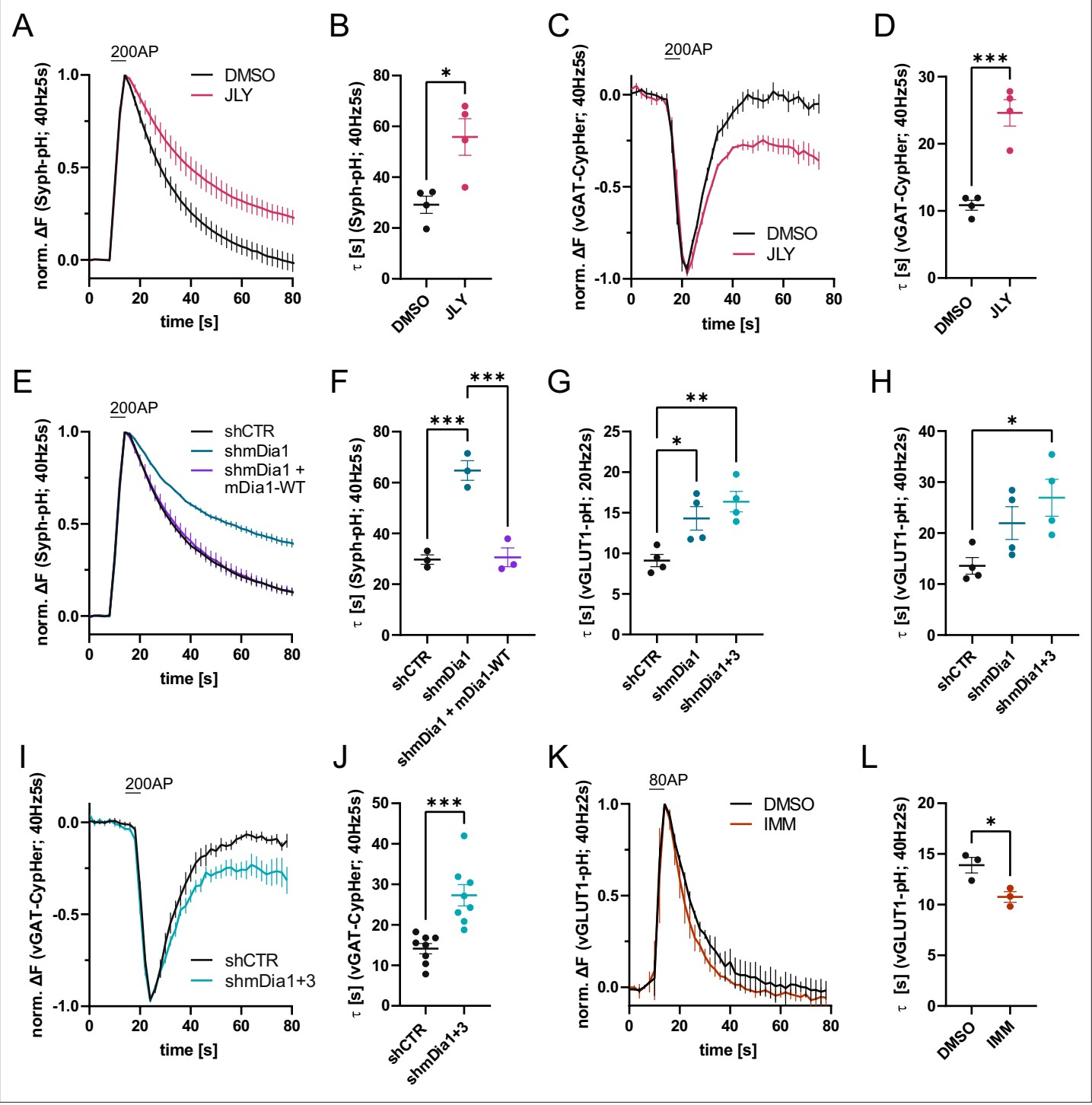

**Figure 1.** Actin dynamics and actin-nucleating mDia1/3 proteins facilitate synaptic vesicle (SV) endocytosis. (**A**) Averaged normalized Synaptophysin-pHluorin (Syph-pH) fluorescence traces from transfected hippocampal neurons stimulated with 200 action potentials (APs) (40 Hz, 5 s) at physiological temperature (37.5 °C). Neurons were treated with 0.1% dimethyl sulfoxide (DMSO) or JLY cocktail (containing 8 μM Jasplakinolide, 5 μM Latrunculin A, and 10 μM Y-27632) as indicated. Data shown represent the mean ± SEM. N=4 independent experiments from $n_{DMSO}$ = 23 videos; $n_{JLY}$ = 36 videos. (**B**) Endocytic decay constants ($\tau$) of Synaptophysin-pHluorin traces in A: $\tau_{DMSO}$ = 29.1±3.4 s; $\tau_{JLY}$ = 55.8±7.2 s; p<0.05, two-tailed student's t-test. Data shown represent mean ± SEM. (**C**) Averaged normalized bleach-corrected vGAT-CypHer fluorescence traces from hippocampal neurons treated with DMSO or JLY cocktail in response to 200 AP (40 Hz, 5 s) stimulation. Data shown represent the mean ± SEM. N=4 independent experiments from $n_{DMSO}$ = 23 videos; $n_{JLY}$ = 29 videos. (**D**) Endocytic decay constants of vGAT-CypHer traces in C: $\tau_{DMSO}$ = 10.9±0.7 s, $\tau_{JLY}$ = 24.6±2.0 s; p<0.001, two-tailed unpaired student's t-test. Data shown represent the mean ± SEM. (**E**) Averaged normalized Synaptophysin-pHluorin fluorescence traces from

*Figure 1 continued on next page*

*Figure 1 continued*

hippocampal neurons transfected with shRNA-encoding plasmids against no mammalian target (*shCTR*) or *Diaph1* (*shmDia1*) in response to 200 AP (40 Hz, 5 s) stimulation. Neurons were co-transfected with mDia1-mCherry (mDia1-WT) or mCherry alone (*shCTR* & *shmDia1*) to exclude artifacts from overexpression. Data shown represent the mean ± SEM. N=3 independent experiments from $n_{shCTR}$ = 28 videos, $n_{shmDia1}$=21 videos, $n_{shmDia1 + mDia1-WT}$=21 videos. (**F**) Endocytic decay constants of Synaptophysin-pHluorin traces in E: $\tau_{shCTR}$ = 29.7±1.9 s; $\tau_{shmDia1}$ = 64.7 ± 3.9 s; $\tau_{shmDia1 + mDia1-WT}$ = 30.6±3.7 s; $p_{shCTR\ vs\ shmDia1}$ <0.001, $p_{shmDia1\ vs\ shmDia1 + mDia1-WT}$<0.001, one-way ANOVA with Tukey's post-test. Data shown represent mean ± SEM. (**G**) Endocytic decay constants of averaged normalized vesicular glutamate transporter 1 (vGLUT1)-pHluorin fluorescence traces (*Figure 1—figure supplement 1I*) from hippocampal neurons transduced with *shCTR* ($\tau_{shCTR}$ = 9.1±0.8 s), *shmDia1* ($\tau_{shmDia1}$ = 14.3±1.5 s), or *shmDia1 +3* ($\tau_{shmDia1+3}$ = 16.4±1.3 s) in response to 40 AP (20 Hz, 2 s) stimulation ($p_{shCTR\ vs\ shmDia1}$ <0.05, $p_{shCTR\ vs\ shmDia1+3}$ < 0.01, one-way ANOVA with Tukey's post-test). Data shown represent mean ± SEM. N=4 independent experiments from $n_{shCTR}$ = 17 videos; $n_{shmDia1}$=19 videos; $n_{shmDia1+3}$ = 18 videos. (**H**) Endocytic decay constants of averaged normalized vGLUT1-pHluorin fluorescence traces (*Figure 1—figure supplement 1K*) of neurons transduced with lentiviral vectors encoding *shCTR* ($\tau_{shCTR}$ = 13.6±1.6 s), *shmDia1* ($\tau_{shmDia1}$ = 22.0±3.2 s) or *shmDia1 +3* ($\tau_{shmDia1+3}$ = 26.9±3.6 s) in response to 80 AP (40 Hz, 2 s) stimulation ($p_{shCTR\ vs\ shmDia1+3}$ < 0.05, one-way ANOVA with Tukey's post-test). Data shown represent mean ± SEM. N=4 independent experiments from $n_{shCTR}$ = 12 videos, $n_{shmDia1}$=15 videos, $n_{shmDia1+3}$ = 18 videos. (**I**) Averaged normalized bleach-corrected vGAT-CypHer fluorescence traces from hippocampal neurons transduced with *shCTR* or *shmDia1 +3* in response to 200 AP (40 Hz, 5 s) stimulation. Data shown represent the mean ± SEM. N=8 independent experiments from $n_{shCTR}$ = 37 videos, $n_{shmDia1+3}$ = 35 videos. (**J**) Endocytic decay constants of vesicular γ aminobutyric acid transporter (vGAT)-CypHer traces in I: $\tau_{shCTR}$ = 14.1±1.3 s; $\tau_{shmDia1+3}$ = 27.3±2.6 s; p<0.001, two-tailed unpaired student's t-test. Data shown represent the mean ± SEM. (**K**) Averaged normalized vGLUT1-pHluorin fluorescence traces from transduced neurons in response to 80 AP (40 Hz, 2 s) stimulation. Cells were treated with 0.1% DMSO or 10 μM mDia activator (IMM) in the imaging buffer. Data shown represent mean ± SEM. N=3 independent experiments from $n_{DMSO}$ = 18 videos; $n_{IMM}$ = 16 videos. (**L**) Endocytic decay constants of vGLUT1-pHluorin traces in K: $\tau_{DMSO}$ = 14.9±0.8 s; $\tau_{IMM}$ = 9.8±0.5 s; p<0.05, two-tailed unpaired student's t-test. Data shown represent mean ± SEM.

The online version of this article includes the following source data and figure supplement(s) for figure 1:

**Source data 1.** Numerical source data for *Figure 1A–L*.

**Figure supplement 1.** Role of formins and mDia1/3 in synaptic vesicle (SV) endocytosis.

**Figure supplement 1—source data 1.** Numerical source data of *Figure 1—figure supplement 1A*, B, C, D, E, F, G, I, J, K, L, M, O.

**Figure supplement 1—source data 2.** Original scan for the anti-mDia1 immunoblot from *Figure 1—figure supplement 1H*.

**Figure supplement 1—source data 3.** Original scan for the anti-mDia3 immunoblot from *Figure 1—figure supplement 1H*.

**Figure supplement 1—source data 4.** Original scan for the anti-tubulin immunoblot from *Figure 1—figure supplement 1H*.

**Figure supplement 1—source data 5.** Original scans for immunoblots from *Figure 1—figure supplement 1H* with highlighted bands and sample labels.

in apparent SV fusion (*Figure 1—figure supplement 1O*). Ultrastructural analysis of hippocampal synapses from mDia1/3-depleted neurons by electron microscopy revealed a reduction in SV density compared to controls (*Figure 2A and B*). This phenotype was rescued upon prior silencing of endogenous neuronal network activity in the presence of tetrodotoxin (TTX) (*Figure 2C* and *Figure 2—figure supplement 1A and B*). A similar, slightly more pronounced depletion of SVs was observed in hippocampal synapses from *Diaph1* knockout (mDia1KO) mice (*Peng et al., 2007*; *Peng et al., 2003*; *Shinohara et al., 2012*; *Figure 2D and E*).

These data are consistent with a role for actin dynamics (see *Figure 1*) and actin nucleating mDia1/3 proteins in presynaptic endocytosis and SV recycling.

## mDia facilitates SV endocytosis by regulating presynaptic F-actin

mDia1 in addition to its Rho binding domain and a C-terminal actin-assembling formin homology 2 (FH2) domain comprises an unstructured N-terminal region involved in membrane binding (*Eisenmann et al., 2005*; *Ramalingam et al., 2010*; *Figure 3A*). This architecture suggests that mDia specifically promotes the nucleation of unbranched actin filaments at membranes, for example, to promote endocytosis at the presynaptic plasma membrane. To test this hypothesis, we assayed the ability of mutant mDia1 lacking its N-terminal membrane-binding region (mDia1-ΔN) (*Ramalingam et al., 2010*) to rescue defective SV endocytosis in hippocampal neurons depleted of endogenous mDia1. Mutant mDia1-ΔN displayed reduced association with membranes in transfected HEK293T cells (*Figure 3—figure supplement 1A and B*) and was unable to restore normal kinetics of Syph-pHluorin endocytosis in hippocampal neurons transfected with *Diaph1*-specific microRNA (shmDia1miR; *Figure 3B* and *Figure 3—figure supplement 1C*). Hence, mDia may act on SV endocytosis by specifically promoting actin assembly at presynaptic membranes. We probed this model at several levels. We analyzed the nanoscale localization and distribution of mDia1 via multicolor time-gated

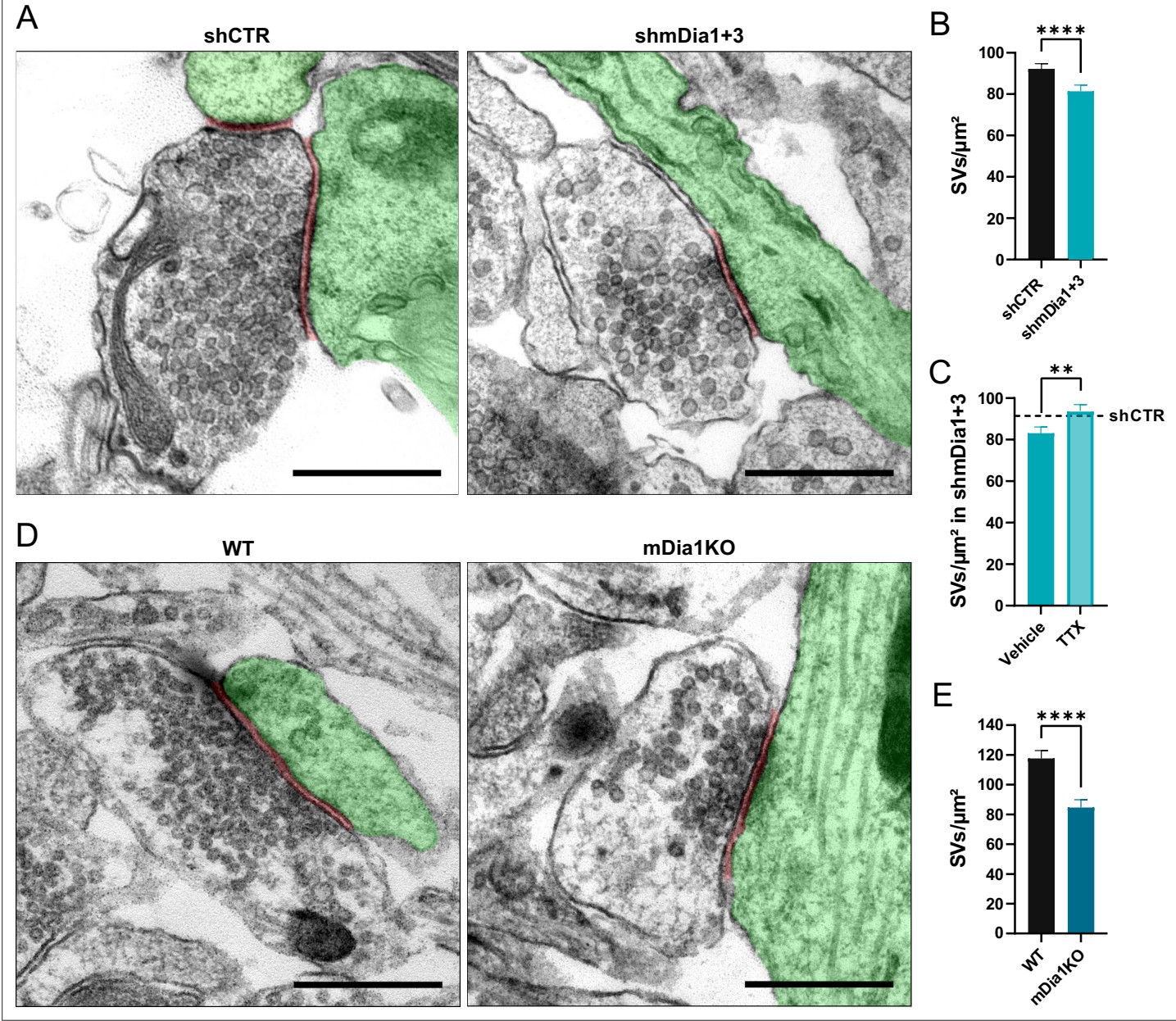

**Figure 2.** Loss of mDia1/3 causes an activity-dependent reduction of the synaptic vesicle (SV) pool. (**A**) Representative synaptic electron micrographs from hippocampal neurons transduced with lentiviruses encoding *shCTR* or *shmDia1 +3*, targeting *Diaph1/2* genes. Postsynapse and postsynaptic cleft are colored in green and maroon, respectively. Scale bar, 250 nm. (**B**) Average number of SVs per µm² in boutons from *shCTR* (92.2±2.5) and *shmDia1 +3* (81.4±2.9; p<0.0001, Mann-Whitney test) treated neurons. Data shown represent mean ± SEM. N=3 independent experiments from $n_{shCTR}$ = 326 synapses, $n_{shmDia1+3}$ = 321 synapses. (**C**) Average number of SVs per µm² in synaptic boutons from hippocampal neurons transduced with lentiviruses encoding *shmDia1 +3* and treated with 0.1% Vehicle (10 µM NaOAc; 83.2±2.9) or 1 µM Tetrodotoxin (TTX; 93.8±3.1; p<0.01, Mann-Whitney test) for 36 hr before chemical fixation. Data shown represent mean ± SEM from two independent experiments and $n_{Vehicle}$ = 225 synapses, $n_{TTX}$ = 221 synapses. Representative synaptic electron micrographs are shown in *Figure 2—figure supplement 1A*. The dotted line represents the average SV numbers/µm² in *shCTR* boutons treated with dimethyl sulfoxide (DMSO) from the same experiments as a reference (*Figure 2—figure supplement 1A*), (**B**). (**D**) Representative electron micrographs of synapses in hippocampal neurons from wild-type (WT) or *Diaph1* (encoding mDia1) knockout (KO) mice. Postsynapse and postsynaptic cleft are colored in green and maroon, respectively. Scale bar, 250 nm. (**E**) Average number of SVs per µm² in WT (117.6±5.3) and *mDia1* KO (84.6±5.2; p<0.0001, Mann-Whitney test) boutons. Data shown represent the mean ± SEM from $n_{WT}$ = 103, $n_{mDia1KO}$ = 96 synapses (N=1).

The online version of this article includes the following source data and figure supplement(s) for figure 2:

**Source data 1.** Numerical source data for *Figure 2C, D and G*.

*Figure 2 continued on next page*

*Figure 2 continued*

**Figure supplement 1.** Synaptic vesicle (SV) depletion in mDia1/3-depleted neurons is activity-dependent.

**Figure supplement 1—source data 1.** Numerical source data of *Figure 2—figure supplement 1B*.

stimulated emission depletion (gSTED) microscopy. Hippocampal neurons were stained for mDia1, the presynaptic active zone marker Bassoon, and post-synaptic Homer1. The distributions of all three proteins along line profiles perpendicular to the synaptic cleft (i.e. the space between Bassoon and Homer1 clusters) were analyzed (*Gerth et al., 2017*; *Figure 3—figure supplement 1H*). This analysis revealed that endogenous mDia1 is primarily localized to the presynaptic compartment, where it is concentrated at or very close to the plasma membrane (*Figure 3C and D*), i.e., <50 nm away from the presynaptic membrane see *Dani et al., 2010*. Prompted by these results we dissected the interactome of mDia1 by unbiased quantitative proteomic analyses of immunoprecipitates from detergent-extracted mouse synaptosomes (*Figure 3—figure supplement 1D*). In addition to actin, we identified a variety of additional actin-regulatory proteins including Myosin IIB (*Chandrasekar et al., 2013*) and the small GTPase Rac1 (*Figure 3E*). We confirmed the association of mDia1 with actin and Myosin IIB (*Figure 3—figure supplement 1E*) and found Myosin IIB to be concentrated at or near the presynaptic plasma membrane (*Figure 3—figure supplement 1F and G*) akin to the localization of mDia1. The association of mDia1 with Myosin IIB is consistent with the inhibitory effects of the Myosin II inhibitor blebbistatin on SV endocytosis (*Chandrasekar et al., 2013*; *Soykan et al., 2017*). Interestingly, we also observed an enrichment of several endocytic proteins previously implicated in SV recycling in mDia1 immunoprecipitates. These include the lipid phosphatase Synaptojanin1, the bin-amphiphysin-rvs (BAR) domain proteins Amphiphysin, Endophilin A1, and PACSIN1/2 and, the GTPase Dynamin1 (*Chanaday et al., 2019*; *Gan and Watanabe, 2018*; *Saheki and De Camilli, 2012*; *Soykan et al., 2016*; *Figure 3E*) as confirmed by immunoblotting (*Figure 3F*). Based on the association of mDia1 with endocytic proteins including Dynamin1, we hypothesized that mDia1 might get stalled at endocytic intermediates that accumulate under conditions of perturbed Dynamin function (*Ferguson et al., 2009*; *Raimondi et al., 2011*). Application of the partially non-selective Dynamin inhibitor Dynasore (*Macia et al., 2006*; *Figure 3C and H*) or expression of dominant-negative mutant Dynamin1 K44A (*Figure 3G1*) led to a significant accumulation of mDia1 at presynaptic sites marked by Bassoon. mDia1 levels at the postsynapse were also increased (*Figure 3—figure supplement 1I*). Dynasore did not impact the levels (*Figure 3—figure supplement 1J*) or distribution of Bassoon or Homer1.

Based on these results we next investigated the relationship between mDia1/3 and presynaptic actin. Multicolor gSTED imaging confirmed that F-actin labeled by phalloidin was highly abundant at the postsynaptic compartment. In addition, F-actin was also visible, albeit at reduced levels, in presynaptic boutons marked by Bassoon (*Figure 4A and B*). Inhibition of Dynamin in the presence of Dynasore (*Figure 4C* and *Figure 4—figure supplement 1A*) or expression of Dynamin1 K44A (*Figure 4—figure supplement 1B and C*) significantly elevated presynaptic F-actin levels within the Bassoon area. These data are consistent with a role of presynaptic actin in SV endocytosis. We directly tested the role of mDia in presynaptic actin nucleation and actin-dependent SV endocytosis in several ways. First, we generated a mutant version of mDia1 that lacks the ability to nucleate F-actin due to site-specific inactivation of its FH2 domain (*Daou et al., 2014*; *Higashi et al., 2008*; *Xu et al., 2004*). We found that actin polymerization-defective mutant mDia1 (K994A) failed to restore delayed vGLUT1-pHluorin endocytosis in mDia1-depleted hippocampal neurons (*Figure 4D* and *Figure 4—figure supplement 1D*). Second, we analyzed the effects of depleting hippocampal neurons of mDia1/3 on presynaptic F-actin. We found that loss of mDia1/3 significantly reduced the amount of detectable F-actin within the Bassoon area (*Figure 4A and E*). To unequivocally distinguish between presynaptic actin at exo-/ endocytic sites and the abundant pool of postsynaptic actin associated with the postsynaptic density, we used the ORANGE strategy for CRISPR-based genome-engineering (*Willems et al., 2020*) to generate eGFP-*Actb*–knock-in (eGFP-β-actin KI) hippocampal neurons. Due to the low knock-in efficiency, this strategy allowed us to visualize endogenously tagged eGFP-β-actin by 2-color gSTED imaging exclusively in presynaptic boutons marked by vGLUT1. This assay confirmed the significant reduction in endogenous β-actin at presynapses from mDia1/3-depleted neurons (*Figure 4F and G*). Third, we reasoned that if loss of presynaptic actin in mDia-depleted neurons was causal for the observed defect in SV endocytosis, pharmacological stabilization of F-actin might rescue the endocytic phenotype.

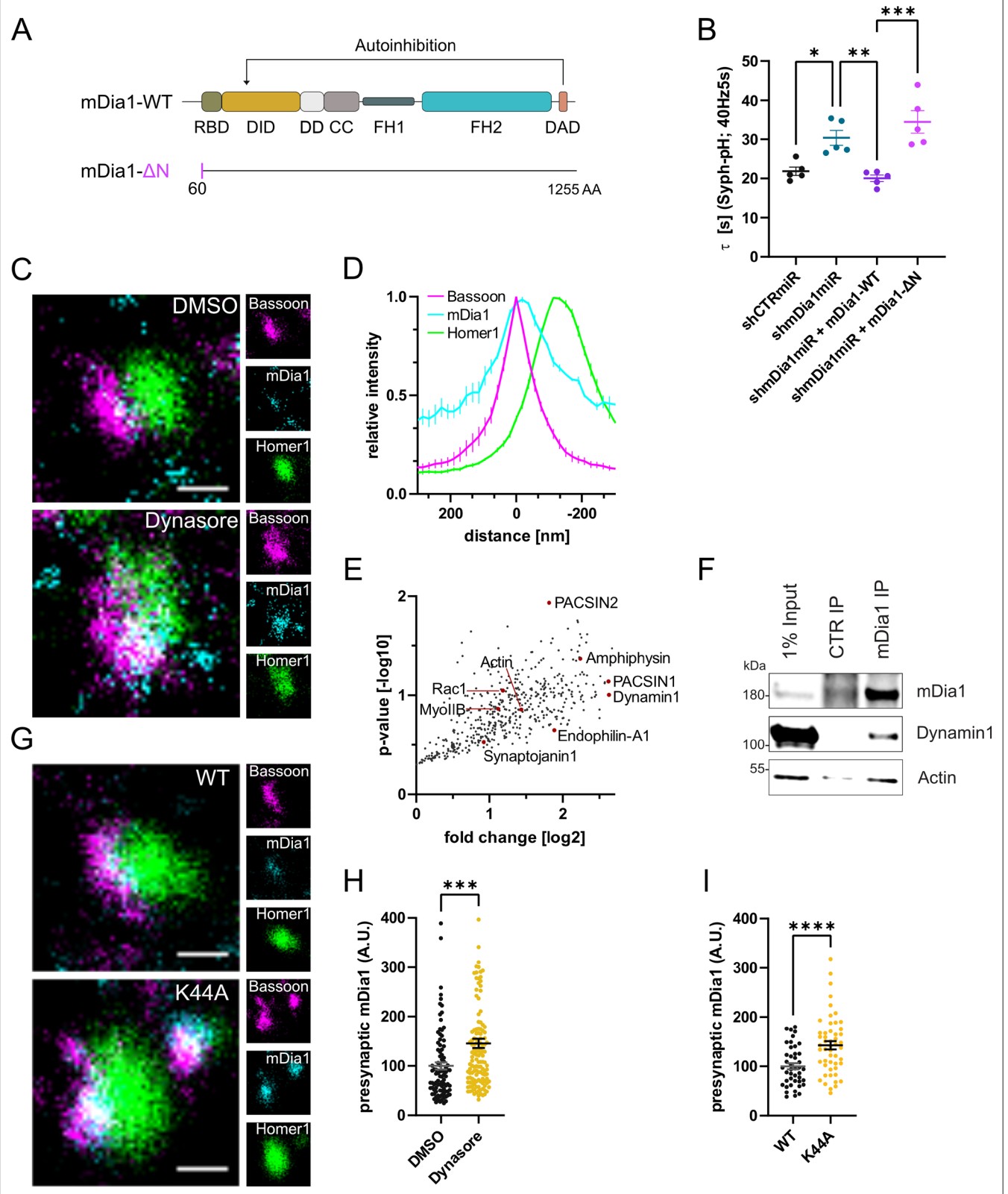

**Figure 3.** mDia1 associates with endocytic proteins and localizes to presynaptic sites. (**A**) Schematic representation of functional domains of mDia1. Rho-binding domain (RBD), Diaphanous inhibitory domain (DID), Dimerization domain (DD), Coiled coil domain (CC), Formin homology domain 1 (FH1), Formin homology domain 2 (FH2), Diaphanous autoinhibitory domain (DAD). The unstructured N-terminus (first 60 amino acids) contains three basic stretches and was truncated in the ΔN mutant. (**B**) Endocytic decay constants of Synaptophysin-pHluorin traces (*Figure 3—figure supplement 1C*)

*Figure 3 continued on next page*

*Figure 3 continued*

from hippocampal neurons transfected with shRNAmiR against no mammalian target (*shCTRmiR*) or *Diaph1* (*shmDia1miR*) in response to 200 action potential (AP) (40 Hz, 5 s) stimulation. For rescue experiments, neurons were co-transfected with plasmids encoding mDia1-WT-mCherry ($\tau_{shmDia1miR + mDia1-WT}$=20.0±0.8 s), mDia1-ΔN-mCherry ($\tau_{shmDia1miR + mDia1-ΔN}$=34.5±2.9 s), or mCherry alone ($\tau_{shCTRmiR}$ = 21.8±1.1 s, $\tau_{shmDia1miR}$ = 30.4±1.9 s) to exclude artifacts from overexpression ($p_{shCTRmiR\ vs\ shmDia1miR}$ < 0.05; $p_{shmDia1miR\ vs\ shmDia1miR + mDia1-WT}$<0.01; $p_{shmDia1miR + mDia1-WT\ vs\ shmDia1miR + mDia1-ΔN}$<0.01, one-way ANOVA with Tukey's post-test). Data shown represent mean ± SEM. N=5 independent experiments from $n_{shCTRmiR}$ = 41 videos, $n_{shmDia1miR}$ = 51 videos, $n_{shmDia1miR + mDia1-WT}$=35 videos, $n_{shmDia1miR + mDia1-ΔN}$=37 videos. (**C**) Representative three-channel time-gated stimulated emission depletion (STED) images of synapses from hippocampal cultures treated with 0.1% dimethyl sulfoxide (DMSO) or 80 µM Dynasore for 10 min before fixation and immunostained for Bassoon (presynaptic marker, magenta), mDia1 (cyan) and Homer1 (postsynaptic marker, green). Scale bar, 250 nm. (**D**) Averaged normalized line profiles for synaptic distribution of mDia1 and Homer1 relative to Bassoon (Maximum set to 0 nm). Data represent mean ± SEM. N=3 independent experiments from n=235 synapses. (**E**) Volcano plot of proteins associating with synaptic mDia1 analyzed by label-free proteomics of anti-mDia1 versus control (CTR) immunoprecipitates from detergent-extracted mouse synaptosomes (P2' fraction). The logarithmic ratios of protein intensities are plotted against negative logarithmic p-values derived from a two-tailed student's t-test. N=3 independent experiments. Each dot represents one protein. Selected cytoskeletal hits include: Actin, Myosin IIB (MyoIIB), and Rac1. Selected endocytic hits include Amphiphysin (p<0.05), Dynamin1, Endophilin-A1, PACSIN1, PACSIN2 (p<0.05), and Synaptojanin1. (**F**) Endogenous immunoprecipitation of mDia1 from detergent-extracted mouse synaptosomes (P2' fraction) using mDia1-specific antibodies. Immunoprecipitates were analyzed by immunoblotting for mDia1, Dynamin1, and β-Actin. (**G**) Representative three-channel time-gated STED images of synapses from hippocampal cultures transduced with wild-type Dynamin1 (WT) or GTPase-deficient Dynamin1 (K44A) in response to 200 AP (40 Hz, 5 s) stimulation. Cells were immunostained for Bassoon (magenta), mDia1 (cyan), and Homer1 (green). Scale bar, 250 nm. (**H**) Presynaptic mDia1 levels in synapses treated with 0.1% DMSO (100±7.3) or 80 µM Dynasore (145.8±9.3; p=0.0001; one sample Wilcoxon test) for 10 min in response to 200 AP (40 Hz, 5 s) stimulation. Absolute line profiles of mDia1 overlapping with Bassoon (presynapse) distribution were integrated. Data shown are normalized to DMSO (set to 100) and expressed as mean ± SEM. N=3 independent experiments from $n_{DMSO}$ = 92 synapses, $n_{Dynasore}$ = 135 synapses. (**I**) Presynaptic mDia1 levels in synapses from hippocampal neurons transduced with wild-type Dynamin1 (WT; 100±6.2) or GTPase-deficient Dynamin1 (K44A; 142.9±8.3, p<0.0001, one sample Wilcoxon test) in response to 200 AP (40 Hz, 5 s) stimulation. Line profiles of mDia1 overlapping with Bassoon distribution were integrated. Data shown are normalized to Dynamin1-WT (set to 100) and expressed as mean ± SEM. N=2 independent experiments from $n_{WT}$ = 43 synapses, $n_{K44A}$ = 51 synapses.

The online version of this article includes the following source data and figure supplement(s) for figure 3:

**Source data 1.** Numerical source data for *Figure 3B, D, E, H and K*.

**Source data 2.** Original scan for the anti-mDia1 and anti-Dynamin1 immunoblots from *Figure 3F*.

**Source data 3.** Original scan for the anti-Actin immunoblot from *Figure 3F*.

**Source data 4.** Original scans for immunoblots from *Figure 3F* with highlighted bands and sample labels.

**Figure supplement 1.** mDia1 binds membranes and localizes to presynaptic endocytic sites.

**Figure supplement 1—source data 1.** Original scan for in-gel mCherry fluorescence from *Figure 3—figure supplement 1A*.

**Figure supplement 1—source data 2.** Original scan for the anti-LAMP1 immunoblot from *Figure 3—figure supplement 1A*.

**Figure supplement 1—source data 3.** Original scans from *Figure 3—figure supplement 1A* with highlighted bands and sample labels.

**Figure supplement 1—source data 4.** Numerical source data of *Figure 3—figure supplement 1B*, C, D, G, I, J.

**Figure supplement 1—source data 5.** Original scans for mCherry fluorescence in gels used for analysis are shown in *Figure 3—figure supplement 1B*.

**Figure supplement 1—source data 6.** Original scans for mCherry fluorescence in gels used for analysis are shown in *Figure 3—figure supplement 1B* with highlighted bands and sample labels.

**Figure supplement 1—source data 7.** Original scan for the anti-mDia1 immunoblot from *Figure 3—figure supplement 1E*.

**Figure supplement 1—source data 8.** Original scan for the anti-Myosin IIB immunoblot from *Figure 3—figure supplement 1E*.

**Figure supplement 1—source data 9.** Original scan for the anti-Actin immunoblot from *Figure 3—figure supplement 1E*.

**Figure supplement 1—source data 10.** Original scans for immunoblots in *Figure 3—figure supplement 1E* with highlighted bands and sample labels.

Treatment of mDia1/3-depleted neurons with jasplakinolide indeed fully restored the kinetics of SV endocytosis monitored by vGLUT1-pHluorin (*Figure 4H* and *Figure 4—figure supplement 1H*) and rescued presynaptic F-actin levels to that of controls (actin levels normalized to shCTR + DMSO, set to 100: shCTR + DMSO = 100 ± 6.3; shmDia1+3 + DMSO=47.7 ± 4.3; shCTR + Jasp = 150.6 ± 11.9; shmDia1+3 + Jasp=94.3 ± 11.5) (*Figure 4—figure supplement 1G*).

We conclude that mDia1/3 facilitates SV endocytosis by regulating presynaptic F-actin.

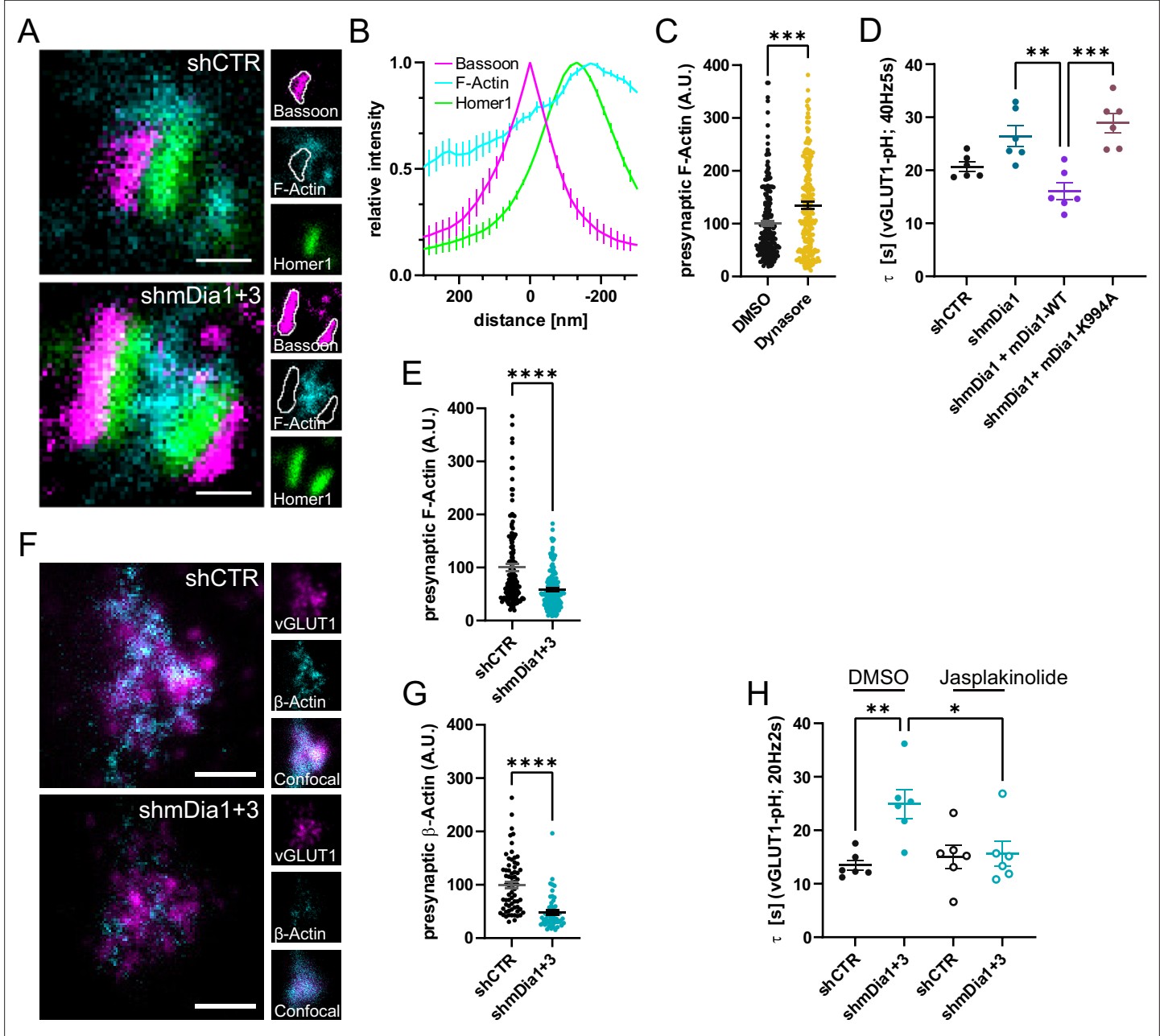

**Figure 4.** mDia facilitates synaptic vesicle (SV) endocytosis by regulating presynaptic F-actin. (**A**) Representative three-channel time-gated stimulated emission depletion (STED) images of synapses from hippocampal cultures transduced with *shCTR* or *shmDia1 +3*, targeting *Diaph1/2* genes, fixed and immunostained for Bassoon (presynaptic marker, magenta), F-Actin (cyan) and Homer1 (postsynaptic marker, green). Scale bar, 250 nm. (**B**) Averaged normalized line profiles for synaptic distribution of F-Actin and Homer1 relative to Bassoon (Maximum set to 0 nm). Data are expressed as mean ± SEM. N=4 independent experiments from n=154 synapses. (**C**) Presynaptic F-Actin levels in synapses treated with 0.1% dimethyl sulfoxide (DMSO) (100±4.8) or 80 μM Dynasore (134.7±6.8; p=0.001, one sample Wilcoxon test) for 10 min before fixation (Representative images in *Figure 4—figure supplement 1A*). Cells were immunostained for Bassoon (magenta), F-Actin (cyan), and Homer1 (green). Absolute line profiles of F-Actin overlapping with Bassoon (presynapse) distribution were integrated. Data shown are normalized to DMSO (set to 100) and expressed as mean ± SEM. N=3 independent experiments from $n_{DMSO}$ = 207 synapses, $n_{Dynasore}$ = 211 synapses. (**D**) Endocytic decay constants of vesicular glutamate transporter 1 (vGLUT1)-pHluorin traces from hippocampal neurons transduced with lentiviral particles encoding *shCTR* ( $\tau_{shCTR}$ = 20.7 ± 0.9 s) or *shmDia1* ( $\tau_{shmDia1}$ = 26.4±2.0 s) in response to 200 action potential (AP) (40 Hz, 5 s) stimulation. For rescue experiments, neurons were co-transduced with lentiviruses encoding mDia1-WT-SNAP ( $\tau_{shmDia1 + mDia1-WT}$=16.1±1.9 s) or mDia1-K994A-SNAP ( $\tau_{shmDia1 + mDia1-K994A}$=29.0±1.9 s) ($p_{shmDia1 \, vs \, shmDia1 + mDia1-WT}$<0.01; $p_{shmDia1 + mDia1-WT \, vs \, shmDia1 + mDia1-K994A}$<0.001, one-way ANOVA with Tukey's post-test). Data are expressed as mean ± SEM. N=6 independent experiments from $n_{shCTR}$ = 21 videos; $n_{shmDia1}$=21 videos, $n_{shmDia1 + mDia1-WT}$=16 videos, $n_{shmDia1 + mDia1-K994A}$=19 videos. (**E**) Presynaptic F-Actin levels in synapses from hippocampal cultures transduced with *shCTR* (100±6.4) or *shmDia1 +3* (58.1±2.9; p<0.001, one sample Wilcoxon test). Line profiles of F-Actin overlapping with

*Figure 4 continued on next page*

*Figure 4 continued*

Bassoon (presynapse) distribution were integrated. Data shown are normalized to shCTR (set to 100) and expressed as mean ± SEM. N=4 independent experiments from $n_{shCTR}$ = 155 synapses, $n_{shmDia1+3}$ = 158 synapses. (**F**) Representative confocal and two-channel time-gated STED images of endogenous β-Actin (cyan) in vGLUT1 (magenta) positive boutons from hippocampal neurons transduced with lentiviruses encoding *shCTR* or *shmDia1 +3*. Scale bar, 250 nm. (**G**) Analysis of presynaptic endogenous β-Actin levels in vGLUT1 positive boutons from *shCTR* (100±6.3) and *shmDia1 +3* (47.7±4.3; p<0.0001 one-sample Wilcoxon test) transduced neurons. β-Actin STED mean intensity was measured using a confocal vGLUT1 signal as a mask. Data shown are normalized to shCTR (set to 100) and expressed as mean ± SEM from two independent experiments and $n_{shCTR}$ = 67 synapses, $n_{shmDia1+3}$ = 53 synapses. (**H**) Endocytic decay constants of vGLUT1-pHluorin traces (*Figure 4—figure supplement 1H*) for neurons transduced with *shCTR* or *shmDia1 +3* in response to 40 AP (20 Hz, 2 s) stimulation. Neurons were pre-incubated with 0.1% DMSO or 1 μM Jasplakinolide (Jasp) for 30 min in the media before imaging ($\tau_{shCTR + DMSO}$ = 13.4±1.0 s, $\tau_{shCTR + Jasp}$ = 15.0±2.2 s, $\tau_{shmDia1+3 + DMSO}$=25.0±2.7 s, $\tau_{shmDia1+3 + Jasp}$=15.6±2.4 s; $p_{shCTR \, vs \, shmDia1+3}$ < 0.01; $p_{shmDia1+3 + DMSO \, vs \, shmDia1+3 + Jasp}$<0.05, one-way ANOVA with Tukey's post-test). Data are expressed as mean ± SEM. N=6 independent experiments from $n_{shCTR + DMSO}$ = 32 videos, $n_{shmDia1+3 + DMSO}$=35 videos, $n_{shCTR + Jasp}$ = 33 videos; $n_{shmDia1+3 + Jasp}$=34 videos.

The online version of this article includes the following source data and figure supplement(s) for figure 4:

**Source data 1.** Numerical source data for *Figure 4B, C, D, E, G and H*.

**Figure supplement 1.** mDia1 regulates presynaptic actin and synaptic vesicle (SV) endocytosis.

**Figure supplement 1—source data 1.** Numerical source data of *Figure 4—figure supplement 1C*, D, E, F, H.

## mDia1/3-Rho and Rac1 signaling pathways cooperatively act to facilitate presynaptic endocytosis

In non-neuronal cells, mDia proteins have been shown to be disinhibited by active Rho family GTPases, in particular RhoA (*Otomo et al., 2005*; *Figure 5A* and *Figure 1—figure supplement 1N*). We reasoned that Rho proteins might serve a similar function at presynaptic nerve terminals in hippocampal neurons. Multicolor gSTED imaging revealed that endogenous RhoA was mostly located within the presynaptic compartment (*Figure 5B and C*). Interference with Rho function by co-expression of dominant-negative (DN) variants of RhoA and RhoB (*Figure 5D* and *Figure 5—figure supplement 1A*) or co-depletion of these Rho isoforms, by shRNA-mediated targeting of *Rhoa* and *Rhob* genes (shRhoA +B; *Figure 5—figure supplement 1C*), delayed SV endocytosis in response to stimulation with 200 APs without impacting the apparent levels of exocytosis (*Figure 5—figure supplement 1B and D*). These results are consistent with a model in which active Rho promotes mDia function and, thereby F-actin nucleation.

F-actin dynamics are known to be controlled by interdependent signaling networks characterized by feedback regulation between components within the same pathway and between key regulatory switches (e.g. individual Rho/Rac family members) that drive distinct forms of F-actin (see below) (*Lawson and Ridley, 2018*). For example, active mDia is known to stimulate RhoA activation (*Kitzing et al., 2007*; *Figure 5A*). Consistently, we found that RhoA activity was reduced in hippocampal neurons depleted of mDia1/3 (*Figure 5E and F*). Work in non-neuronal cells has further revealed that RhoA activity antagonizes activation of Rac1 (*Chauhan et al., 2011*), a key factor for promoting the formation of branched F-actin networks at the cell cortex (*Hodge and Ridley, 2016*; *Lawson and Ridley, 2018*; *Figure 6A*). We hypothesized that reciprocally interdependent Rho/ Rac1 signaling might control presynaptic F-actin assembly and, thereby, SV endocytosis. In agreement with the alleged antagonistic regulation of RhoA and Rac1 function (*Lawson and Ridley, 2018*), we found Rac1 activity to be significantly elevated in hippocampal neurons depleted of mDia1/3 (*Figure 6B and C*), e.g., under conditions of reduced Rho-GTP levels. A similar increase in active Rac1 levels was observed upon pharmacological inhibition of RhoA/B in the presence of Rhosin (*Figure 6—figure supplement 1A*). Multicolor gSTED imaging showed that Rac1 was equally distributed between pre- and postsynaptic compartments in unperturbed hippocampal neurons (*Figure 6D and E*). Elevated active Rac1 levels might conceivably ameliorate presynaptic endocytic phenotypes elicted by mDia1/3 loss. In support of this hypothesis, we found that selective pharmacological inhibition of Rac1 reduced presynaptic actin levels (*Figure 6—figure supplement 1B and C*) and caused a delay in the endocytic retrieval of endogenous vGAT (*Figure 6F and G*). Moreover, Rac1 inhibition appeared to further aggravate impaired vGAT endocytosis in mDia1/3-depleted neurons, although the effect remained below statistical significance. Expression of constitutively active GTP-locked Rac1 (Rac1-CA) restored normal Syph-pHluorin endocytosis kinetics in mDia1/3-depleted neurons, whereas Syph-pHluorin endocytosis was delayed upon expression of dominant-negative Rac1 (Rac1-DN) and exacerbated the endocytic phenotype of mDia1/3 loss (*Figure 6H* and *Figure 6—figure supplement 1E*). No

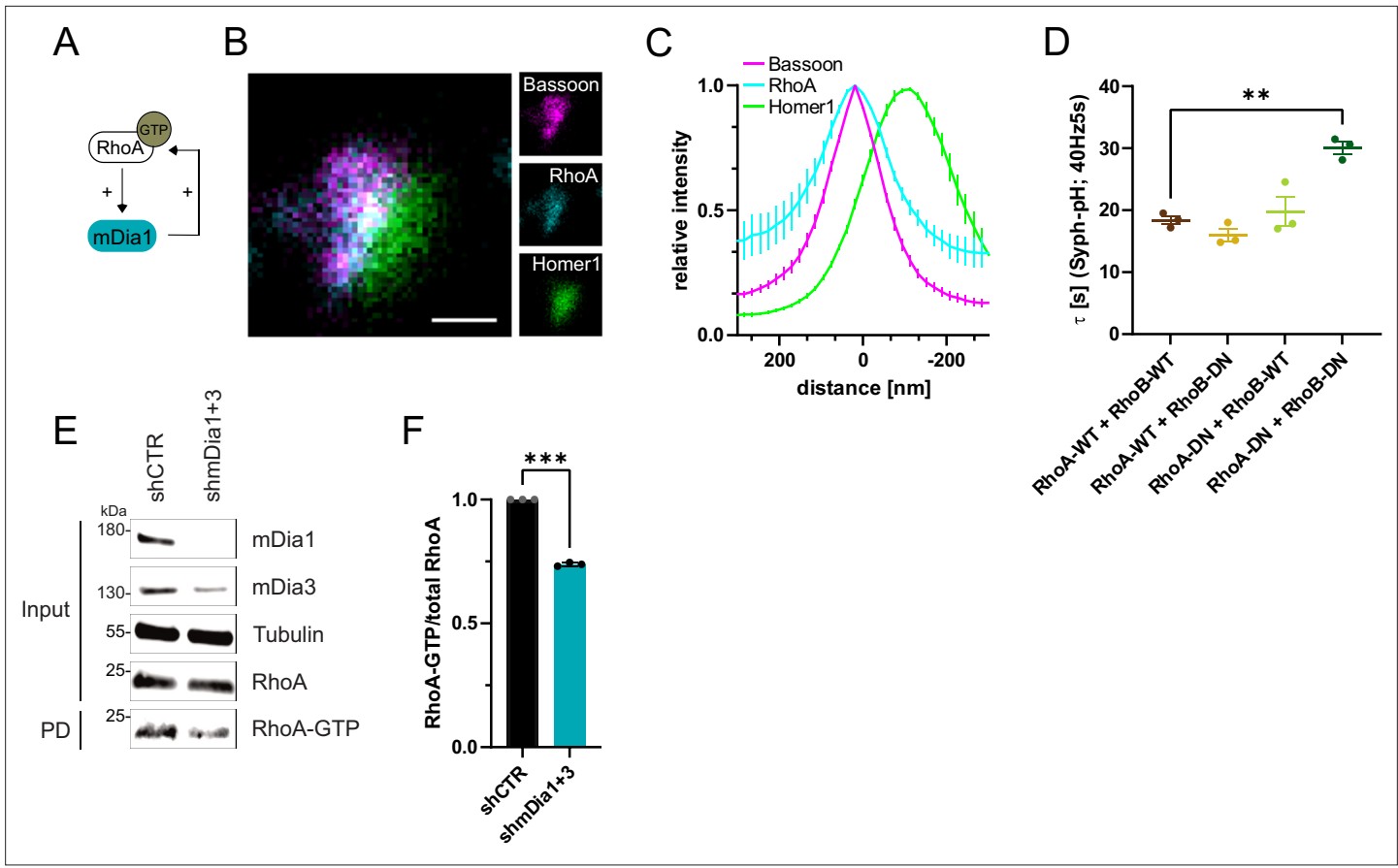

**Figure 5.** RhoA/B facilitates presynaptic endocytosis and are regulated by mDia1/3. (**A**) Schematic representation of activation of mDia1 by RhoA-GTP and positive feedback loop of mDia1 on RhoA-GTP levels through GEF stimulation. (**B**) Representative three-channel time-gated STED image of synapses from hippocampal cultures, fixed and immunostained for Bassoon (magenta), RhoA (cyan), and Homer1 (green). Scale bar, 250 nm. (**C**) Averaged normalized line profiles for synaptic distribution of RhoA and Homer1 relative to Bassoon (Maximum set to 0 nm). Data are expressed as mean ± SEM. N=5 independent experiments from n=230 synapses. (**D**) Endocytic decay constants of averaged normalized Synaptophysin-pHluorin fluorescence traces (*Figure 5—figure supplement 1A*) in response to 200 action potential (AP) (40 Hz, 5 s) stimulation. Neurons were transfected with the annotated combinations of plasmids encoding wild-type (WT) or dominant-negative (DN, T19N mutation) RhoA and RhoB ($\tau_{\text{RhoA-WT + RhoB-WT}}$=18.4±0.7 s, $\tau_{\text{RhoA-WT + RhoB-DN}}$ = 16.0±1.0 s, $\tau_{\text{RhoA-DN + RhoB-WT}}$=19.8±2.4 s, $\tau_{\text{RhoA-DN + RhoB-DN}}$=30.1±1.0 s; $p_{\text{RhoA-WT + RhoB-WT vs RhoA-DN + RhoB-DN}}$<0.01, one-way ANOVA with Tukey's post-test). Data shown represent mean ± SEM. N=3 independent experiments from $n_{\text{RhoA-WT + RhoB-WT}}$=21 videos, $n_{\text{RhoA-DN + RhoB-WT}}$=31 videos, $n_{\text{RhoA-WT + RhoB-DN}}$=23 videos, $n_{\text{RhoA-DN + RhoB-DN}}$=22 videos. (**E**) Analysis of RhoA activity by RhoA-GTP pulldown (PD) from whole-cell lysates (input) of mouse hippocampal neurons expressing shCTR or shmDia1 +3 using immobilized Rhotekin as a bait. Samples were analyzed by immunoblotting for mDia1, mDia3, RhoA, and Tubulin using specific antibodies. Input, 10% of material used for the pulldown. The contrast of pulldown and input blots was seperately adjusted for visualization purposes. (**F**) Densitometric quantification of RhoA-GTP normalized to total RhoA levels (input) in lysates from neurons transduced with shCTR or shmDia1 +3 (0.7±0.0, p<0.001, one sample t-test) from immunoblots exemplified in E. Values for shCTR were set to 1. Data are expressed as mean ± SEM from N=3 independent experiments.

The online version of this article includes the following source data and figure supplement(s) for figure 5:

**Source data 1.** Numerical source data for *Figure 5C, D and F*.

**Source data 2.** Original scans for the anti-mDia1, anti-Tubulin, anti-RhoA, and anti-mDia3 immunoblots from *Figure 5E*.

**Source data 3.** Original scans for immunoblots from *Figure 5E* with highlighted bands and sample labels.

**Source data 4.** Original scans for the anti-RhoA immunoblots used for analysis are shown in *Figure 5F*.

**Source data 5.** Original scans for immunoblots used for analysis are shown in *Figure 5F* with highlighted bands and sample labels.

**Figure supplement 1.** RhoA/B regulates synaptic vesicle (SV) endocytosis.

**Figure supplement 1—source data 1.** Numerical source data of *Figure 5—figure supplement 1A–D*.

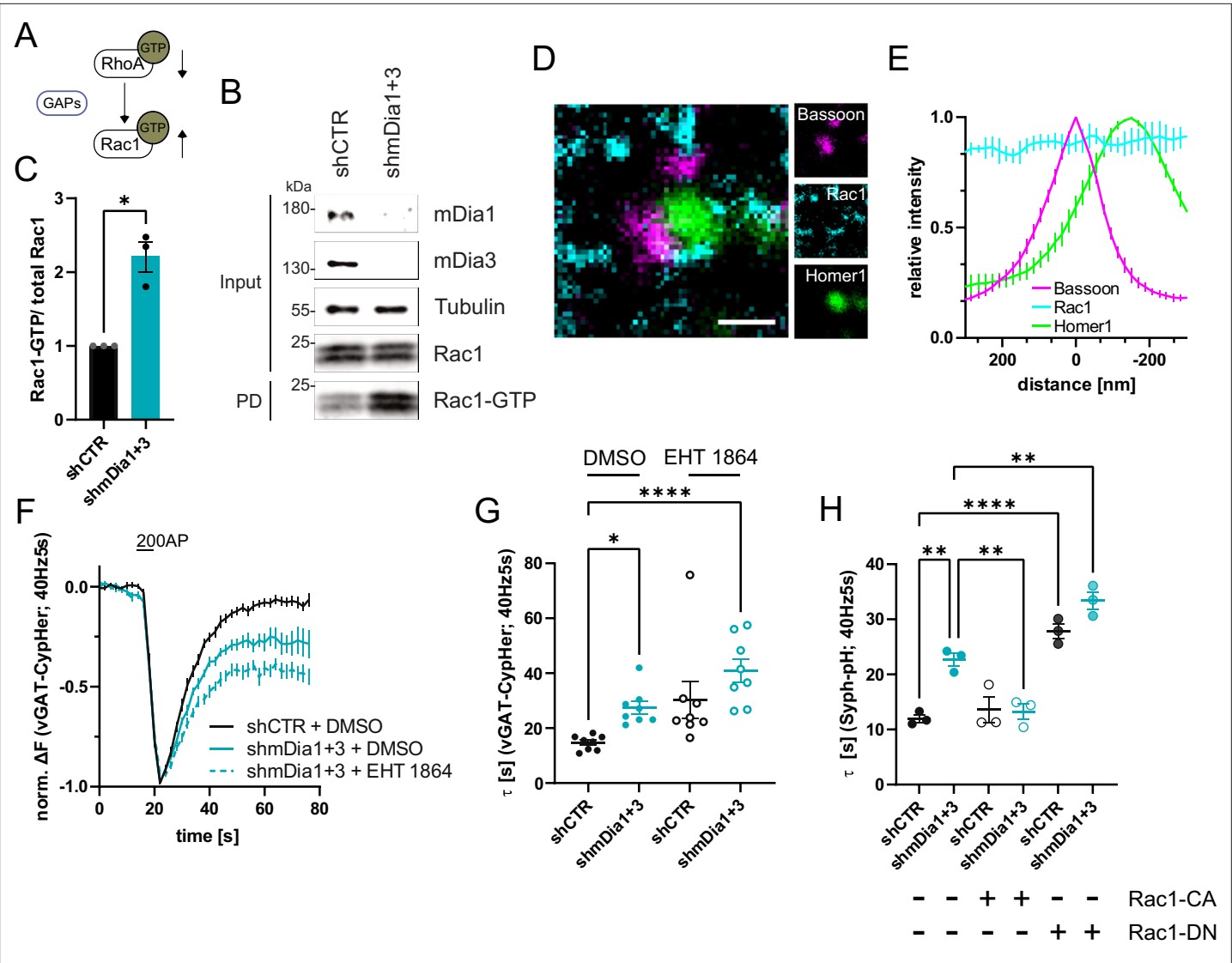

**Figure 6.** mDia1/3-Rho and Rac1 signaling facilitates presynaptic endocytosis. (**A**) Schematic of the interplay between RhoA and Rac1 signaling via GTPase regulatory proteins (e.g. GTPase activating proteins (GAPs) among others) common for RhoA and Rac1. (**B**) Analysis of Rac1 activity by Rac1-GTP pulldown (PD) from whole-cell lysates (input) of mouse hippocampal neurons expressing shCTR or shmDia1 +3 utilizing immobilized PAK as a bait. Samples were analyzed by immunoblotting for mDia1, mDia3, Rac1, and Tubulin using specific antibodies. Input, 10% of material used for the pulldown. The contrast of pulldown and input blots was seperately adjusted for visualization purposes. (**C**) Densitometric quantification of Rac1-GTP normalized to total Rac1 levels (input) in lysates from neurons transduced with shCTR or shmDia1 +3 (2.2±0.2; p<0.05, one sample t-test) from immunoblots exemplified in (**B**). Values for shCTR were set to 1. Data are expressed as mean ± SEM from N=3 independent experiments. (**D**) Representative three-channel time-gated stimulated emission depletion (STED) image of synapses from hippocampal cultures, fixed and immunostained for Bassoon (magenta), Rac1 (cyan), and Homer1 (green). Scale bar, 250 nm. (**E**) Averaged normalized line profiles for synaptic distribution of Rac1 and Homer1 relative to Bassoon (Maximum set to 0 nm). Data represent mean ± SEM. N=3 independent experiments from n=79 synapses. (**F**) Averaged normalized vGAT-CypHer fluorescence traces for neurons transduced with *shCTR* or *shmDia1 +3* in response to 200 AP (40 Hz, 5 s) stimulation. Cells were acutely treated with 0.1% DMSO or 10 µM Rac1 Inhibitor (EHT 1864) in the imaging buffer. Data shown represent the mean ± SEM. N=8 independent experiments from $n_{shCTR + DMSO}$ = 46 videos, $n_{shmDia1+3 + DMSO}$ = 45 videos, $n_{shCTR + EHT 1864}$ = 42 videos, $n_{shmDia1+3 + EHT 1864}$ = 43 videos. (**G**) Endocytic decay constants of vGAT-CypHer traces in F: $\tau_{shCTR + DMSO}$ = 14.7±0.9 s, $\tau_{shmDia1+3 + DMSO}$=27.5±2.3 s, $\tau_{shCTR + EHT 1864}$ = 30.3±6.7 s, $\tau_{shmDia1+3 + EHT 1864}$ = 41.0±4.3 s; $p_{shCTR + DMSO \, vs \, shmDia1+3 + DMSO}$<0.05, $p_{shCTR + DMSO \, vs \, shmDia1+3 + EHT 1864}$ < 0.0001, Kruskal-Wallis test with Dunn's post-test. Data represent mean ± SEM. (**H**) Endocytic decay constants of Synaptophysin-pHluorin traces (***Figure 6—figure supplement 1E***) of neurons transduced with *shCTR* ($\tau_{shCTR}$ = 12.0±0.7 s) or *shmDia1 +3* ($\tau_{shmDia1+3}$ = 22.7±2.0 s) and transfected with constitutively active Rac1 (Rac1-CA; Q61L variant; $\tau_{shCTR + Rac1-CA}$=13.6±1.2 s, $\tau_{shmDia1+3 + Rac1-CA}$=13.3±1.4 s) or dominant negative Rac1 (Rac1-DN; T17N variant; $\tau_{shCTR + Rac1-DN}$ = 27.8±1.3 s, $\tau_{shmDia1+3 + Rac1-DN}$ = 33.4±1.6 s) in response to 200 AP (40 Hz, 5 s) stimulation ($p_{shCTR \, vs \, shmDia1+3}$ < 0.01; $p_{shCTR \, vs \, shCTR + Rac1-DN}$<0.0001, $p_{shCTR \, vs \, shmDia1+3 + Rac1-DN}$<0.01, $p_{shmDia1+3 \, vs \, shmDia1+3 + Rac1-DN}$<0.01, one-way ANOVA with Tukey's

*Figure 6 continued on next page*

*Figure 6 continued*

post-test). Data are expressed as mean ± SEM. N=3 independent experiments from $n_{shCTR}$ = 12 videos, $n_{shmDia1+3}$ = 23 videos; $n_{shCTR + Rac1-CA}$=10 videos, $n_{shmDia1+3 + Rac1-CA}$=14 videos, $n_{shCTR + Rac1-DN}$ = 9 videos; $n_{shmDia1+3 + Rac1-DN}$ = 13 videos.

The online version of this article includes the following source data and figure supplement(s) for figure 6:

**Source data 1.** Original scans for the anti-mDia3, anti-Tubulin, and anti-Rac1 immunoblots from *Figure 6B*.

**Source data 2.** Original scan for the anti-mDia1 immunoblot from *Figure 6B*.

**Source data 3.** Original scans for immunoblots from *Figure 6B* with highlighted bands and sample labels.

**Source data 4.** Numerical source data for *Figure 6C, E, F, G and H*.

**Source data 5.** Original scans for the anti-Rac1 immunoblots used for analysis are shown in *Figure 6C*.

**Source data 6.** Original scans for immunoblots used for analysis are shown in *Figure 6C* with highlighted bands and sample labels.

**Figure supplement 1.** Cooperative action of mDia1/3 and Rac1 pathways in presynaptic endocytosis.

**Figure supplement 1—source data 1.** Original scan for the anti-Rac1 immunoblots from *Figure 6—figure supplement 1A*.

**Figure supplement 1—source data 2.** Original scan for the anti-Tubulin immunoblot from *Figure 6—figure supplement 1A*.

**Figure supplement 1—source data 3.** Original scans for immunoblots in *Figure 6—figure supplement 1A* with highlighted bands and sample labels.

**Figure supplement 1—source data 4.** Numerical source data of *Figure 6—figure supplement 1C*, D, E, F, G, I, J, K.

**Figure supplement 1—source data 5.** Original scans for anti-Cdc42 immunoblots used for analysis are shown in *Figure 6—figure supplement 1G*.

**Figure supplement 1—source data 6.** Original scans for anti-Cdc42 immunoblots used for analysis are shown in *Figure 6—figure supplement 1G* with highlighted bands and sample labels.

---

overt effect of Rac1 inhibition or overexpression of either Rac1 form on SV exocytosis was observed (*Figure 6—figure supplement 1D and F*). Perturbation of the related Cdc42 protein, another actin regulatory GTPase *Lawson and Ridley, 2018* found at synapses (*Figure 6—figure supplement 1G–I*), did not significantly affect the kinetics of SV endocytosis in control or mDia1/3-depleted neurons (*Figure 6—figure supplement 1J and K*). These data suggest that interdependent mDia1/3 and Rac1-based pathways control SV endocytosis via presynaptic actin.

We probed this model by analyzing the ultrastructure of synapses from hippocampal neurons depleted of mDia1/3 or following inhibition of Rac1 activity. Synapses from mDia1/3-depleted neurons displayed an accumulation of non-coated plasma membrane invaginations that were found both in the vicinity of the active zone as well as distal from the synaptic contact area (*Figure 7A and B*). Moreover, we observed elevated numbers of endosome-like vacuoles (ELVs) (*Figure 7A and C*) that might serve as a donor membrane for SV reformation (*Kononenko and Haucke, 2015*; *Kononenko et al., 2014*; *Watanabe et al., 2014*) as suggested by the observed reduction in SV numbers in mDia-depleted neurons (compare *Figure 2*). A prominent accumulation of plasma membrane invaginations (*Figure 7—figure supplement 1A*) and endosome-like vacuoles (*Figure 7—figure supplement 1B*) was similarly found in hippocampal neurons from mDia1 KO mice. Inhibition of Rac1 function by EHT 1864 also led to the accumulation of non-coated plasma membrane invaginations (*Figure 7A and D*), whereas the number of endosome-like vacuoles was not significantly altered (*Figure 7A and E*). Pharmacological blockade of Rac1 activity in neurons depleted of mDia1/3 further increased the number of plasma membrane invaginations significantly, while the additional effect on the number of endosome-like vacuoles remained insignificant (*Figure 7F, G and H*). mDia1/3 loss and Rac1 perturbation exhibited similar phenotypes with respect to the length and width of tubular membrane invaginations (*Figure 7—figure supplement 1C and D*).

Collectively, these findings demonstrate that mDia1/3-Rho and Rac1 signaling pathways cooperatively act to facilitate presynaptic endocytosis and SV recycling.

## Discussion

We found that loss of mDia1 either alone or in conjunction with the closely related mDia3 isoform slows the kinetics of SV endocytosis of exogenous pHluorin-tagged SV proteins and of endogenous vGAT without perturbing exocytic SV fusion. These phenotypes are accompanied by the accumulation of plasma membrane invaginations and vacuolar structures as well as an activity-dependent reduction of the total SV pool (*Figures 1, 2 and 7*). Furthermore, we observe that interdependent mDia1/3-Rho and Rac1 signaling pathways cooperatively act to facilitate presynaptic endocytosis

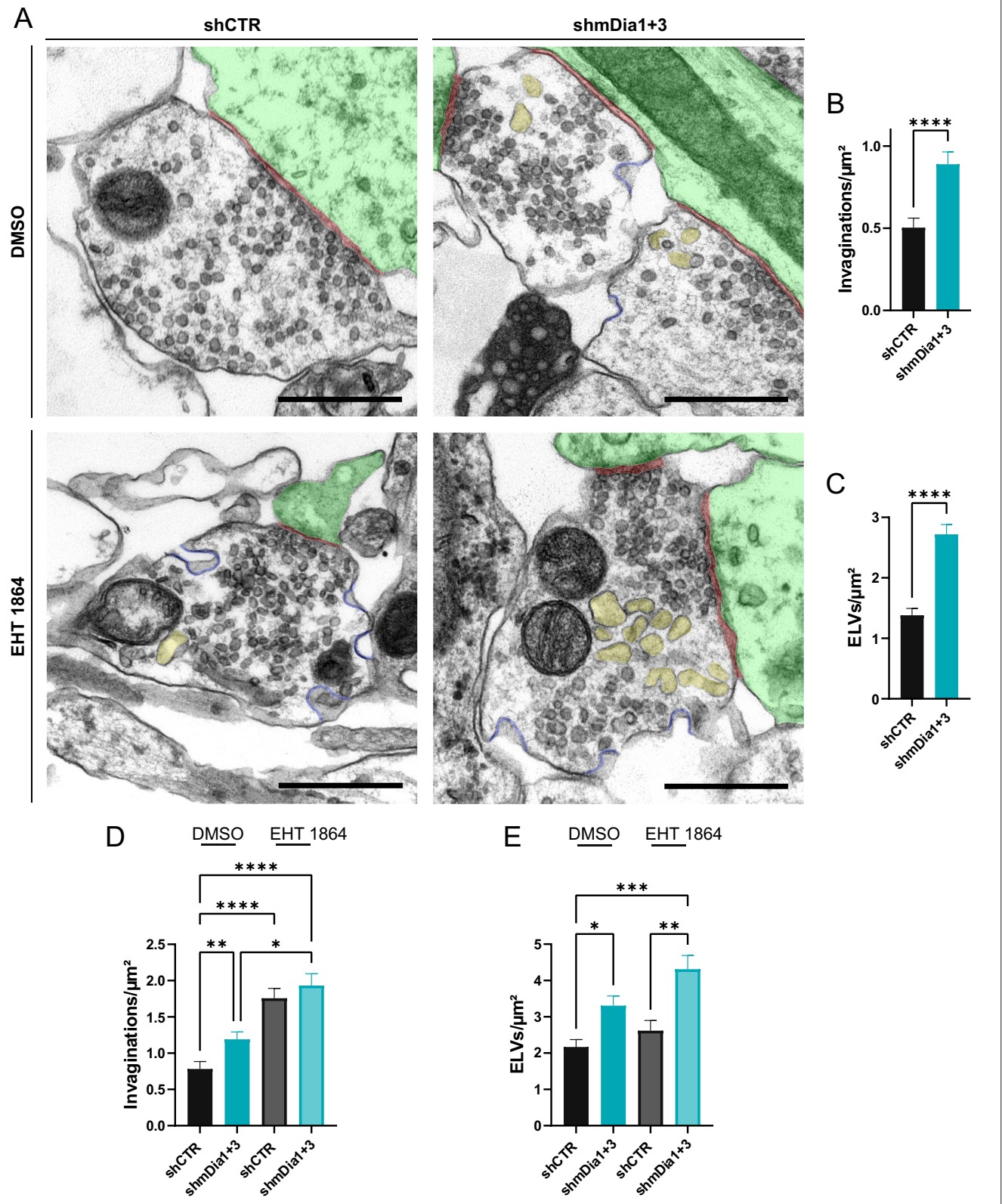

**Figure 7.** Defects in presynaptic ultrastructure induced by interference with mDia1/3-Rho and Rac1 signaling. (**A**) Representative synaptic electron micrographs from hippocampal neurons transduced with lentiviruses encoding *shCTR* or *shmDia1 +3*, targeting *Diaph1/2* genes, and treated with 0.1% dimethyl sulfoxide (DMSO) or 10 μM Rac1 Inhibitor (EHT 1864) for 2 hr before fixation. Invaginations and endosome-like vacuoles (ELVs) are colored in blue and yellow, while postsynapse and synaptic cleft are colored in green and maroon, respectively. Scale bar, 250 nm. (**B**) Average number

*Figure 7 continued on next page*

*Figure 7 continued*

of invaginations per µm$^2$ in *shCTR* (0.5±0.1) and *shmDia1 +3* (0.9±0.1; p<0.0001, Mann-Whitney test) boutons. Data represent mean ± SEM. N=3 independent experiments from $n_{shCTR}$ = 326 synapses, $n_{shmDia1+3}$ = 323 synapses. (**C**) Average number of ELVs per µm$^2$ in shCTR (1.4±0.1) and *shmDia1 +3* (2.7±0.2; p<0.0001, Mann-Whitney test) boutons. Data represent mean ± SEM. N=3 independent experiments from $n_{shCTR}$ = 326 synapses, $n_{shmDia1+3}$ = 323 synapses. (**D**) Average number of invaginations per µm$^2$ in *shCTR* and *shmDia1 +3* boutons treated with 0.1% DMSO (0.8±0.1 for *shCTR*; 1.2±0.1 for *shmDia1 +3*, $p_{shCTR + DMSO \text{ vs } shmDia1+3 + DMSO}$<0.01) or 10 µM EHT 1864 (1.8±0.1 for *shCTR*, $p_{shCTR + DMSO \text{ vs } shCTR + EHT 1864}$ < 0.0001; 1.9±0.2 for *shmDia1 +3*, $p_{shCTR + DMSO \text{ vs } shmDia1+3 + EHT 1864}$ < 0.0001, $p_{shmDia1+3 + DMSO \text{ vs } shmDia1+3 + EHT 1864}$ < 0.05, Kruskal-Wallis test with Dunn's post-test) for 2 hr before fixation. Data represent mean ± SEM from $n_{shCTR + DMSO}$ = 144 synapses, $n_{shmDia1+3 + DMSO}$=143 synapses, $n_{shCTR + EHT 1864}$ = 136 synapses, $n_{shmDia1+3 + EHT 1864}$ = 153 synapses. (**E**) Average number of ELVs per µm$^2$ in *shCTR* and *shmDia1 +3* boutons treated with 0.1% DMSO (2.2±0.2 for *shCTR*; 3.3±0.3 for *shmDia1 +3*, $p_{shCTR + DMSO \text{ vs } shmDia1+3 + DMSO}$<0.05) or 10 µM EHT 1864 (2.6±0.3 for *shCTR*; 4.3±0.4 for *shmDia1 +3*, $p_{shCTR + DMSO \text{ vs } shmDia1+3 + EHT 1864}$ < 0.001, $p_{shCTR + EHT 1864 \text{ vs } shmDia1+3 + EHT 1864}$ < 0.01, Kruskal-Wallis test with Dunn's post-test) for 2 hr before fixation. Data represent mean ± SEM from $n_{shCTR + DMSO}$ = 144 synapses, $n_{shmDia1+3 + DMSO}$=143 synapses, $n_{shCTR + EHT 1864}$ = 136 synapses, $n_{shmDia1+3 + EHT 1864}$ = 153 synapses.

The online version of this article includes the following source data and figure supplement(s) for figure 7:

**Source data 1.** Numerical source data from *Figure 7B–E*.

**Figure supplement 1.** mDia1/3 and Rac1 cooperatively regulate the synaptic vesicle (SV) cycle and presynaptic ultrastructure.

**Figure supplement 1—source data 1.** Numerical source data of *Figure 7—figure supplement 1A–D*.

(*Figures 5–7*). Several lines of evidence indicate that these endocytic phenotypes are a consequence of perturbed presynaptic actin levels or dynamics. (i) Endogenous presynaptic F-actin is reduced in mDia1/3-depleted hippocampal synapses (*Figure 4*) and (ii) actin polymerization-defective mDia1 fails to restore normal endocytosis kinetics in hippocampal neurons depleted of mDia1 (*Figure 4*). Conversely, (iii) pharmacological stabilization of F-actin by jasplakinolide (*Figure 4*) or (iv) expression of constitutively-active Rac1 to drive compensatory actin polymerization (for example via other formins and/ or ARP2/3) (*Figure 6*) rescues endocytosis in mDia1/3-KD hippocampal neurons. Moreover, (v) we find mDia1 (*Figure 3*) and F-actin (*Figure 4*) to accumulate at presynapses under conditions of impaired dynamin-dependent endocytosis. Together with the observation that inhibition of F-actin dynamics (*Peng et al., 2011*) in the combined presence of latrunculin A, jasplakinolide, and Y-27632 (*Figure 1*) slows the kinetics of SV endocytosis, our data support the hypothesis that presynaptic actin facilitates SV endocytosis downstream of mDia/ RhoA and Rac1-based signaling pathways.

Our data extend previous studies using conditional genetics that have identified an actin requirement for SV endocytosis (*Wu et al., 2016*). The endocytic phenotype described here for loss of mDia1/3, however, is substantially milder than that elicited by complete loss of β− or γ-actin, likely reflecting the fact that several partially redundant pathways cooperate to promote F-actin assembly at synapses (see below). In contrast to *Actb* or *Actg1* KO hippocampal neurons (*Wu et al., 2016*) we did not observe strong exocytic depression in response to train stimulation in mDia-depleted neurons (e.g. *Figure 1—figure supplement 1A–M*) although we cannot rule out mild differences in synaptic strength and/ or release probability as observed in *Rac1* KO hippocampal neurons and at the calyx of Held (*Keine et al., 2022*; *O'Neil et al., 2021*).

At the ultrastructural level, we find that genetic (KO) or lentiviral induced (KD) loss of the genes *Diaph1* or *Diaph1/2*, encoding mDia1 or mDia1/3, respectively, partially phenocopies acute perturbation of formin-mediated actin assembly (*Soykan et al., 2017*) with respect to the accumulation of plasma membrane invaginations. Unlike acute formin inhibition (*Soykan et al., 2017*), we find elevated numbers of post-fission endosome-like vacuoles in mDia-depleted hippocampal synapses. The latter correlates with a mild reduction of SV numbers, suggesting that mDia may have a further role in SV reformation downstream of the actual endocytosis reaction (*Chanaday et al., 2019*; *Gan and Watanabe, 2018*; *Kononenko et al., 2014*). The identified function of mDia1/3-mediated presynaptic actin assembly in presynaptic endocytosis and SV recycling is consistent with the actin-binding properties of many endocytic proteins including Dynamin (*Ferguson et al., 2009*; *Gu et al., 2014*), PACSINs (*Kessels and Qualmann, 2006*), or Epsins, among others (*McMahon and Boucrot, 2011*; *Merrifield et al., 2005*; *Saheki and De Camilli, 2012*). How exactly actin functions to facilitate endocytosis of SV membranes remains to be fully understood. The comparably small size of the membrane invaginations that accumulate following pharmacological inhibition of formins (<100 nm) or Rac1 or upon loss of mDia1 (typically <150 nm, although larger ones are observed) render an actomyosin-based constriction mechanism for SV endocytosis unlikely (consistent with *Sankaranarayanan et al., 2003*). We rather favor a - possibly indirect - function of F-actin in presynaptic endocytosis via the

formation of an F-actin ring that couples exocytic membrane compression to endocytic pit formation (*Ogunmowo et al., 2023*). Such a model is supported by the abundance of large plasma membrane invaginations distal from the active zone where SV fusion occurs at synapses following mDia1/3 and Rac1 loss of function (*Figure 7*). Consistent with this, it has been shown that mDia1 controls presynaptic membrane contractility (*Deguchi et al., 2016*), e.g., via the recently identified presynaptic actin corrals visualized in genome-engineered rat hippocampal neurons (*Bingham et al., 2023*).

A key result from our study is the finding that presynaptic endocytosis is regulated by interdependent, yet partially redundant signaling for F-actin assembly downstream of Rho and Rac1, e.g., pathways known to control linear and branched F-actin networks (*Lawson and Ridley, 2018*; *Müller et al., 2020*) that were shown to co-exist within hippocampal presynaptic boutons (*Bingham et al., 2023*). We demonstrate that loss of mDia1/3 or inhibition of Rho causes the hyperactivation of the actin-promoting GTPase Rac1 and that dual loss of mDia1/3 loss and Rac1 function results in synergistic inhibitory effects with respect to presynaptic endocytosis and ultrastructural defects (*Figures 5–7*). These findings suggest that the signaling network that regulates presynaptic actin assembly may have been evolutionarily selected for plasticity and resilience. A resilient network for the control of actin dynamics at the presynapse may also help to explain the partially contradicting results from drug-based manipulations of actin and actin regulatory factors in different models of presynaptic neurotransmission reported in the past (*Bleckert et al., 2012*; *Del Signore et al., 2021*; *Hori et al., 2022*; *Piriya Ananda Babu et al., 2020*; *Richards et al., 2004*; *Sankaranarayanan et al., 2003*; *Shupliakov et al., 2002*; *Wu and Chan, 2022*; *Wu et al., 2016*). Moreover, accumulating evidence indicates that different pools of actin might be differentially amenable to drug treatments (*Bleckert et al., 2012*), e.g., as a consequence of the association of actin filaments with molecules such as tropomyosin (*Gormal et al., 2017*) that render them resistant to drugs, and/ or variations in the fraction of F-actin *vs.* monomeric actin pools (*Higashida et al., 2008*), providing a possible explanation for previously discrepant results.

A number of open questions remain. We predict that presynaptic actin assembly via mDia/Rho and Rac1 facilitates all forms of presynaptic endocytosis that operate on timescales of milliseconds to seconds (*Chanaday and Kavalali, 2018*; *Clayton and Cousin, 2009*; *Watanabe and Boucrot, 2017*) and at many types of synapses (e.g. small central synapses, ribbon synapses, the neuromuscular junction), but further studies will be needed to test this hypothesis. We note that fast endophilin-mediated endocytosis (FEME), a process with resemblance to endocytosis at synapses (*Watanabe and Boucrot, 2017*), has been shown to be differentially regulated by actin and Rho GTPase family members (*Boucrot et al., 2015*; *Renard et al., 2015*). At present we do not understand if and how RhoA/B and Rac1 GTPase activation is nested into the SV cycle and whether these GTPases operate at the same or different nanoscale sites and on the same or different membranes. Future studies will be required to tackle these questions.

## Materials and methods
### Materials
#### Animals
Primary neurons were obtained from either wild-type C57BL/6 J (Charles River, RRID: IMSR_JAX:000664) or *Diaph1* KO mice (*Peng et al., 2007*; *Peng et al., 2003*). All animal experiments were reviewed and approved by the ethics committee of the *Landesamt für Gesundheit und Soziales* (LAGeSo) Berlin or the Committee on the Ethics of Animal Experiments of Columbia University and conducted according to the committees' guidelines (LAGeSo) or the Guide for the Care and Use of Laboratory Animals of the National Institutes of Health (for *Diaph1* KO mice). At the facilities, animal care officers monitored compliance with all regulations. Mice were group-housed under 12/12 hr light/dark cycle with access to food and water ad libitum. Mice from both genders were used and cultures were randomly allocated to experimental groups (e.g. different treatments). Multiple independent experiments using several biological replicates were carried out as indicated in the Figure legends.

#### Antibodies
Antibodies and their working dilutions used in this study are denoted in the Key Resource Table (IB: Immunoblot; IC: Immunocytochemistry; IP: Immunoprecipitation). Antibodies were stored according

to the manufacturer's recommendations. All secondary antibodies are species-specific (highly cross-adsorbed).

## Cell lines

Human embryonic kidney 293T (HEK293T) cells were obtained from the American Type Culture Collection (Cat# CRL-3216; RRID:CVCL_0063). Cells were cultured in Dulbecco's modified Eagle's medium supplemented with glucose (DMEM; 4.5 g/L; Thermo Fisher Scientific) and 10% heat-inactivated fetal bovine serum (FBS; Gibco), penicillin (100 U/ml; Gibco), and streptomycin (100 µg/ml; Gibco) at 37 °C and 5% $CO_2$. Cells were routinely tested for mycoplasma contamination.

## Chemicals

Compounds were dissolved in dimethyl sulfoxide (DMSO), unless indicated otherwise, and diluted 1:1000 to their working concentrations (see Key Resource Table). For acute pharmacological treatment (in pHluorin/CypHer assays), drugs were added to the imaging buffer. For longer incubations, the conditioned cell media was supplemented with the chemicals (see Key Resource Table for incubation time).

For silencing neuronal network activity, sodium channels were inhibited by the addition of tetrodotoxin (TTX; in 10 mM sodium acetate, pH 5.3) to the neuronal culture medium at a day in vitro (DIV) 12 for 36 hr. As a control, cells from the same preparation were treated with equal volumes of 10 mM sodium acetate (annotated as Vehicle).

## Plasmids

All recombinant DNA reagents used for protein expression are listed in the Key Resource Table: Synaptophysin-pHluorin was a kind gift from Dr. L. Lagnado (Univ. of Sussex, UK), vGLUT1-pHluorin was generated in-house by Svenja Bolz as previously described, while vGLUT1-mCherry was kindly provided by Dr. Franck Polleux (New York, USA). HA-tagged RhoA-WT and RhoA-T19N were kind gifts from Dr. Theofilos Papadopoulos (Göttingen, Germany) and myc-tagged RhoB-WT and RhoB-T19N were kindly gifted by Dr. Harry Mellor (Bristol, UK). All other expression vectors were generated for this study. Plasmids based on pEGFP-C1, pmCherry-N1, pCAG, and pcDNA3 utilizing a CMV, or CAG promotor (see Key Resource table) were used for overexpression based on the transfection of neuronal cells, while pFUGW vectors carrying a human Synapsin1 promotor (hSyn1) were used for the generation of lentiviral particles for the transduction of neurons (see Key Resource Table annotated as 'lentiviral plasmid'). In the course of this study several approaches to deplete *Diaph1*, encoding mDia1, have been carried out and are annotated in the Figure legends: For knockdown of *Diaph1*, *Rhoa,* and *Rhob* by transfection, commercially available lentiviral small hairpin RNA (shRNA) vectors based on the pLKO.1 backbone (see Key Resource Table; annotated as 'transfected') were purchased from Sigma-Aldrich. To reduce the amount of DNA needed for transfection to perform pHluorin assays, vectors expressing shRNA embedded into a microRNA (miRNA) context for *Diaph1* together with Synaptophysin-2x-pHluorin as a reporter were cloned based on pRRLsinPPT-emGFP-miR Control, a kind gift of Dr. Peter S. McPherson (Montreal, Canada). Finally, lentiviral vectors for knockdown of *Diaph1* and *Diaph2* (annotated in Key Resource Table as shmDia1 and shmDia3 'transduced') were generated based on the backbone f(U6) sNLS-RFPw shCTR, a kind gift from Prof. Christian Rosenmund (Berlin, Germany). All vectors used for genetic depletion via RNA interference are listed in the Key Resource Table and express gene-specific shRNA under a U6 promotor, that either targets the coding sequence (CDS) or the 3'-untranslated region (3'-UTR).

## Oligonucleotides

Plasmids generated for this study were cloned using oligonucleotides (BioTez GmbH, Berlin, Germany) listed in *Supplementary file 1* (lower case denotes nucleotides that do not anneal to the backbone; underlined nucleotides denote sense and antisense sequences of shRNA).

## Methods

### Generation of expression plasmids

Expression plasmids generated for this study were cloned by PCR amplification (Phusion High-Fidelity DNA polymerase) and restriction enzyme (Thermo Fisher Scientific; Fast digest) digest according to the manufacturers' manual. pSNAP-N1 was generated in-house by Hannes Gonschior by sub-cloning pSNAP$_f$ (New England BioLabs Inc, Cat#N9183S) via PCR and the restriction enzymes *AgeI* and *NotI* into pmCherry-N1. pFUG_hSyn_MCS was cloned by Amirreza Ohadi to enable simple insertion of SNAP-tagged proteins by inserting a multiple cloning site (MCS) on annealed oligonucleotides into a pFUG_hSyn1 backbone (a gift from Christian Rosenmund, Berlin, Germany) cut by *AgeI* and *EcoRI*. To generate mDia1-WT-mCherry, the sequence encoding mDia1 was cut from mDia1-mEmerald-N1 (Addgene, Cat#54157) and pasted into pmCherry-N1 by *AgeI* and *XhoI* digestion. mDia1-WT-SNAP was cloned by cutting out the coding sequence of mDia1 from mDia1-mEmerald-N1 and pasting it into pSNAP-N1 by *AgeI* and *NheI* digest of both vector and insert. To subclone mDia1-SNAP into a lentiviral vector the coding sequence of mDia1-SNAP was pasted from mDia1-WT-SNAP-N1 into pFUG_hSyn1_MCS utilizing common cut sites for *NheI* and *NotI* digest. For the generation of a lentiviral vector expressing Dynamin-WT, the sequence encoding Dynamin1 was first extracted from Dynamin1-pmCherry-N1 (Addgene; Cat#27697) by *EcoRI* and *XmaI* digestion and cloned into pSNAP-N1 with the same enzymes. Subsequently, Dynamin1-SNAP was isolated by *NheI* and *NotI* digest and transformed into pFUG_hSyn1_MCS. The mDia1- ΔN truncation variant was generated by introducing a new start codon on a primer shifted by 60 amino acids (see ***Supplementary file 1***), PCR amplifying the truncated DNA and subcloning the template into pmCherry-N1 by *BglII* and *AgeI* digest. For introducing point mutations in mDia1 (Lysine-994 to Alanine; K994A) and Dynamin1 (Lysine-44 to Alanine; K44A), the Q5 site-directed mutagenesis kit (New England Biolabs Inc; E0552S) was used according to the manufacturer's manual and oligonucleotides listed in ***Supplementary file 1***.

All vectors were confirmed by Sanger sequencing (LGC Genomics, Berlin, Germany) and amplified by self-made chemically competent TOP10 *Escherichia coli* before purifying respective endotoxin-free DNA by 2-propanole precipitation.

### Isolation, culture, and transfection of primary hippocampal neurons

Neuronal hippocampal cell cultures were prepared as described before (***López-Hernández et al., 2022***; ***Soykan et al., 2017***). In short, hippocampi from postnatal mice (p0-p3) were surgically isolated and dissociated into single cells by trypsin (5 g/L, Sigma-Aldrich) digestion. Neurons (100,000 cells/ well of a six-well) were plated onto poly-L-lysine-coated coverslips and grown in modified Eagle medium (MEM; Thermo Fisher) supplemented with 5% FCS and 2% B-27 (Gibco) and maintained at 5% $CO_2$ and 37 °C in humidified incubators. In addition, 2 µM cytosine β-D-arabinofuranoside (AraC) was added to the cell culture media in the first 2 days in vitro (DIV) to limit glial proliferation. For transient protein expression, neurons were transfected on DIV 7–9 utilizing a Calcium phosphate transfection kit (Promega; Cat# E1200): In brief, 1–6 µg plasmid DNA (per well of a six-well plate) were mixed with 250 mM calcium phosphate ($CaCl_2$) in ultrapure nuclease-free water. The resulting solution was added to equal volumes of 2x 4-(2-hydroxyethyl)–1-piperazineethanesulfonic acid buffered saline (2 x HEPES; 100 µL) and incubated at room temperature for 20 min. Resulting precipitates were added dropwise to cells that had been transferred to osmolarity-adjusted Neurobasal-A (NBA; Gibco) media to induce starvation. After incubation at 37 °C and 5% $CO_2$ for 30 min, neurons were washed thrice with osmolarity-adjusted Hank's balanced salt solutions (HBSS; Gibco) and transferred back to their original conditioned media.

### shRNA cloning and lentivirus production

Knockdown of proteins was achieved through RNA interference either by $CaCl_2$ transfection of gene-specific shRNA encoding vectors (pLKO.1; Sigma-Aldrich or shRNAmiR) on DIV 7 or by transduction of cells with lentiviral particles harboring the gene-specific shRNA on DIV 2. The respective method is indicated in the Figure legends as *transfected* or *transduced*. To reduce the amount of DNA needed for transfection and to improve neuronal health, shRNAmiR were expressed from the 3'UTR of Synaptophysin-pHluorin based on ***Ritter et al., 2017***: The reporter protein/synthetic cassette was generated by PCR amplifying Synaptophysin-2x-pHluorin with oligos (***Supplementary file 1***) harboring *XbaI* and *SalI* sites. The eGFP in the RRLsinPPT-eGFP-miRCTR plasmid (a kind gift

from Dr. Peter S. McPherson, McGill University, Canada) was replaced by the PCR product through a similar restriction digest to yield shCTRmiR. miRmDia1 (start position 4892, targeting sequence matches open-reading frame) was designed with BLOCK-iT (Thermo Fisher Scientific) and subcloned into shCTRmiR to yield shmDia1miR following protocol in *Ritter et al., 2017*.

As both transfection strategies were limited by low efficiency, lentiviral knockdown was carried out: Lentiviral particles were based on a shuttle vector (pFUGw) driving the expression of a nuclear-targeted red fluorescent protein (NLS-RFP) under a human Synapsin1 (hSyn1) promotor to monitor infection efficiency in neurons and a scrambled mouse shRNA against Clathrin without any murine targets, which was used as the control virus (f(U6) sNLS-RFPw shCTR). To prevent crosstalk between the NLS-RFP and other mCherry constructs, a similar backbone expressing the blue fluorescent protein (BFP) as a reporter was generated in-house by Klaas Ypermann: The backbone was digested by *XbaI* and *PacI*, the hSyn1 promotor and mTagBFP (Addgene; #Cat 105772) were amplified with oligos (*Supplementary file 1*), gel extracted and assembled utilizing a Gibson master mix (New England BioLabs Inc; Cat#E2611L) to yield f(U6) BFP shCTR. To deplete mDia1, a shRNA sequence based on 5'-GCCTAAATGGTCAAGGAGATA-3' as the sense nucleotide corresponding to the 3'UTR of mouse *Diaph1* (NM_007858.4; Sigma-Aldrich, Cat# TRCN0000108685) was designed as an oligonucleotide with overhangs (*Supplementary file 1*) and annealed into the f(U6) sNLS-RFPw shCTR backbone cut with *BamHI* and *PacI* to yield the vector f(U6) sNLS-RFPw/BFP shmDia1. For depletion of mDia3, a sense sequence based on 5'-GCCCTAATCCAGAATCTTGTA-3' corresponding to nucleotides 2473–2493 of mouse *Diaph2* (NM_172493.2; Sigma-Aldrich, Cat# TRCN0000108782) was used to construct f(U6) sNLS-RFPw/BFP shmDia3 as described above. Production of lentiviral particles was conducted using the second-generation packaging system: In brief, HEK293T were co-transfected with lentiviral shRNA constructs and the packaging plasmids psPAX2 (Addgene; Cat# 12260) and MD2.G (Addgene; Cat# 12259) using $CaCl_2$. After 12 hr, the cell media was replaced. Virus-containing supernatants were collected at 48 and 72 hr after transfection, filtered to remove cell debris and particles were concentrated (30-fold) via low-speed centrifugation (Amicon Ultra-15, Ultracel-100; Merck Millipore; Cat# UFC9100) before aliquoting and storage at –70 °C. For all experiments, an infection rate of over 95% was achieved at DIV 14–16.

## Microscopy
### Live-imaging of SV recycling
For live-imaging of Synaptophysin and vesicular glutamate transporter 1 (vGLUT1) recycling, SV proteins fused to the green-fluorescent protein-based pH-sensitive fluorescent reporter pHluorin at their luminal domain were overexpressed by plasmid transfection (Synaptophysin-pHluorin; DIV 7) or lentiviral transduction (vGLUT1-pHluorin; DIV 2).

For following endogenous vesicular gamma-aminobutyric acid transporter (vGAT) recycling, spontaneously active synaptic boutons were labeled by incubating cells with CypHer5E-conjugated antibodies against the luminal domain of vGAT (1:500 from a 1 mg/ml stock; Synaptic Systems; Cat#131,103CpH) for 2 hr in their respective conditioned culture media at 37 °C and 5% $CO_2$ prior to imaging.

To investigate the kinetics of SV recycling, neurons at DIV 14–16 were placed into an RC-47FSLP stimulation chamber (Warner Instruments) in osmolarity-adjusted imaging buffer [170 mM sodium chloride (NaCl), 20 mM N-Tris(hydroxyl-methyl)-methyl-2-aminoethane-sulphonic acid (TES), 5 mM Glucose, 5 mM sodium hydrogencabonate ($NaHCO_3$), 3.5 mM potassium chloride (KCl), 1.3 mM $CaCl_2$, 1.2 mM sodium sulfate ($Na_2SO_4$), 1.2 mM magnesium chloride ($MgCl_2$), 0.4 mM potassium dihydrogenphosphate ($KH_2PO_4$), 50 μM (*2 R*)-amino-5-phosphonovaleric acid (AP5) and 10 μM 6-cyano-7-nitroquinoxaline-2,3-dione (CNQX); pH 7.4] at 37 °C (Tempcontrol 37–2 digital). Cells were subjected to electrical field-stimulation (MultiStim System-D330; Digimeter Ltd.) with annotated stimulation trains to evoke action potentials (APs) [40 Hz, 5 s (200 APs), 40 Hz, 2 s (80 APs), 20 Hz, 2 s (40 APs); at 100 mA]. Following changes in fluorescence were tracked by an inverted Zeiss Axiovert 200 M microscope, equipped with a 40 x oil-immersion EC Plan Neofluar objective (NA 1.30), an EM-CCD camera (Evolve Delta 512) and a pE-300[white] LED light source (CoolLED). The scanning format was set to 512x512 pixels with 16-bit sampling. eGFP (Excitation: BP470-40; Emission: BP535-50; Zeiss filter set 38) or Cy5 (Excitation: BP640-30; Emission: BP525-50; Zeiss filter set 50) filter sets were used for pHluorin or CypHer assays, respectively. Images were acquired at 0.5 Hz frame rate for 100 s with 50

(Syph-pHluorin) or 100 (vGLUT1-pH, vGAT-CypHer) ms exposure with an electron multiplying gain of 250 operated through Fiji-based MicroManager 4.11 software.

Analysis of responding boutons was performed through custom-written macros to identify regions of interest (ROIs) ( *Voll, 2020*) in an automated manner using SynActJ (*Schmied et al., 2021*). Such analysis averaged fluorescence for each time point in an image series (video) of at least >20 responding boutons and corrected values for background fluorescence yielding raw background-corrected fluorescence (F). The fold increase of fluorescence after stimulation ($F_{max}$) can serve as a measure for exocytic fusion and is calculated by normalizing F by the mean intensity of F before stimulation (Baseline/basal fluorescence ($F_0$)=mean intensity of first five frames for pHluorin; first 10 frames for CypHer) for each time point (surface normalization = $F/F_0$ for pHluorin; $F_0 - F_{min}$ for CypHer). To account for boutons with varying pHluorin expression in one image series and to compare reacidification kinetics, the fluorescence of each time point was subtracted by the basal fluorescence ($F_0$) to yield $\Delta F$ (F-$F_0$), which was then normalized by its peak value ($\Delta F_{max}$). Resulting peak normalized curves ($\Delta F/ \Delta F_{max}$) are annotated as a norm. $\Delta F$. Endocytic decay constants ($\tau$) were calculated by averaging and then fitting the norm. $\Delta F$ ($\Delta F/\Delta F_{max}$) traces of all videos in one condition (N) to a mono-exponential decay curve [$y_0 +A*e^{(-t/\tau)}$] with the constraints of $y_0$=1 and offset = 0 in Prism 9 (GraphPad).

CypHer traces had to be corrected for photobleaching: The decay constant of the bleaching curve was determined by fitting the data points of the first 10 frames (20 s, prior to stimulation) to a mono-exponential decay curve. The corresponding value of the photobleaching curve at a given time *t* was added to the raw fluorescence intensity measured at each corresponding time point to correct for the loss of intensity due to bleaching.

## Immunocytochemistry

For immunostainings, neuronal cultures were chemically fixed on DIV 14–16 using 4% p-formaldehyde (PFA) and 4% sucrose in phosphate-buffered saline (PBS) for 15 min at room temperature (RT). For experiments in *Figure 3C, G–I*, cultures were stimulated with a 40 Hz train for 5 s in an imaging buffer before immediate fixation. After fixation, cells were washed thrice with PBS and incubated with permeabilization buffer (10% normal goat serum, 0.3% Triton X-100 in PBS) for 30 min at RT, followed by primary antibody incubation of proteins of interest in permeabilization buffer at indicated dilutions (see Key Resource Table) at 4 °C overnight. Subsequently, unbound antibodies were removed by three PBS washes while bound antibodies were decorated by corresponding fluorophore-coupled secondary antibodies in permeabilization buffer for 1 hr at RT. For F-Actin staining, Phalloidin-Alexa Fluor 594 (1:1000 stock; AAT Bioquest; Cat# ABD-23158) was added in the secondary incubation step. Finally, neurons were washed thrice with PBS, followed by two washes with ultra-pure water. Coverslips were dried for 2 hr before mounting in ProLong Gold Antifade (Thermo Fisher Scientific; #P36934) on glass slides (Thermo Fisher Scientific; VWR; Cat#630–1985). The slides were cured for at least 72 hr at RT before imaging.

## Multicolor time-gated STED imaging

STED images were acquired from fixed samples by a HC PL APO CS2 100 x oil objective (1.40 NA) on a Leica SP8 TCS STED 3 x microscope (Leica Microsystems) equipped with a pulsed white-light excitation laser (WLL; ~80 ps pulse width, 80 MHz repetition rate; NKT Photonics) and a STED laser for depletion (775 nm). The scanning format was set to 1024 × 1024 pixels, with 8-bit sampling, 4 x line averaging, 4 x frame accumulation, and 6 x optical zoom, yielding a final pixel size of 18.9 nm. Three-color imaging was performed by sequentially exciting following fluorophores (excitation filter = Exf; emission filter = Emf): Atto647N (Exf: 640 nm; Emf: 650–700 nm); Alexa Fluor 594 (Exf: 590 nm; Emf: 600–640 nm) and Atto542 (Exf: 540 nm; Emf: 550–580 nm) operated by the Leica Application Suite X (Leica Microsystems, 2020). For stimulated depletion of the signal, the 775 nm STED laser was applied to all emissions. Detection of the resulting signal was time-gated by 0.3–6 ns to allow enough time for stimulated depletion and collected by two sensitive HyD detectors at appropriate spectral regions distinct from the STED laser wavelength. Settings in independent experiments were similar between conditions to allow quantification of signals. Raw data obtained from three-channel time-gated STED (gSTED) imaging were analyzed with Fiji. For analysis, only synapses oriented in the xy-plane exhibiting a clear separation of Bassoon (presynapse) and Homer1 (postsynapse) clusters were taken into account. To determine synaptic localization and presynaptic levels of proteins of interest, multicolor

line profiles were measured: A line (1.0 μm length, 0.4 μm width) perpendicular to the synaptic cleft (space between Bassoon & Homer1 cluster) was drawn (*Gerth et al., 2017*) and fluorescence intensity of all three channels along this line was measured using a Fiji Macro (Macro_plot_lineprofile_multicolor; Dr. Kees Straatman, University of Leicester, UK). Resulting profiles were aligned to the maximum Bassoon intensity, which was set to 0 nm. For localization analysis, all three profiles were normalized to their maxima, which were set to 1. For quantification of presynaptic protein levels, only the fractions of the non-normalized line profiles of proteins of interest overlapping with the normalized averaged Bassoon distribution (between 151.4 and –37.8 nm; *Figure 3—figure supplement 1H*) were integrated. Intensities were normalized to controls (DMSO, shCTR, WT) which were set to 100.

## Imaging of endogenous β-Actin levels

To visualize endogenous β-Actin levels, eGFP knock-in following the ORANGE method was performed (*Willems et al., 2020*): Neurons were transfected with vGLUT1-mCherry (Franck Polleux) and pOrange GFP-β-Actin KI (Addgene; Cat#131479) by $CaCl_2$ on DIV 2. Directly following transfection, cells were transduced with shCTR or shmDia1 +3 lentiviral particles carrying BFP as a reporter to prevent crosstalk between mCherry and NLS-RFP expression. Neuronal cultures were chemically fixed on DIV 15. After fixation, endogenous GFP and overexpressed vGLUT1-mCherry intensities were enhanced by additional antibody (anti-GFP and anti-RFP; see Key Resource Table) incubation according to the immunocytochemistry protocol described above. Images of neurons were taken on the STED microscope with and without stimulated depletion (confocal) with the same settings as indicated above. For analysis, confocal images were filtered by Gaussian blur (sigma = 2 pixels) and auto-thresholded using Otsu's method. Whenever confocal vGLUT1 signal co-localized with confocal Actin clusters, the area was chosen as a ROI. Actin intensity was then measured in the ROI applied to the STED image. The intensities of β-Actin were normalized to shCTR which was set to 100.

## Transmission electron microscopy

To investigate the synaptic effects of F-actin manipulations at the ultrastructural level, cells on coverslips were chemically fixed with 2% glutaraldehyde in cacodylate buffer (CDB; 0.1 M sodium cacodylate) for 1 hr at RT. Coverslips were washed thrice with CDB before osmification by 1% (w/v) osmium tetroxide ($OsO_4$) and 1.5% (w/v) potassium hexacyanoferrate ($K_3Fe(CN)_6$) in CBD for 1 hr at 4 °C. After postfixation, cells were stained with 1% (w/v) uranyl acetate, dehydrated by methanol gradients and finally embedded by epoxy resin (Sigma Aldrich; Cat# 45359) infiltration. After polymerization (60 °C, 30 hr), coverslips were removed and ultra-thin (70 nm) sections were cut and contrasted with 2% (w/v) uranyl acetate and 80 mM lead citrate. Eight-bit images were obtained on a Zeiss 900 transmission electron microscope equipped with Olympus MegaViewIII or Olympus Morada G2 digital cameras at 30,000x magnification yielding a pixel size of 1.07 nm. Subsequently, morphometry (density of synaptic vesicles, endosome-like vacuoles, clathrin-coated vesicles, clathrin-coated pits, and non-coated invaginations) was analyzed from synaptic profiles with clearly distinguishable active zones and adjacent synaptic vesicles in a blinded manner.

## Biochemistry
### Immunoblotting

To compare protein levels between experimental conditions, immunoblotting was performed: Protein concentrations in lysates were measured by BCA assay (Thermo Fisher Scientific; Cat#23227) and equal protein amounts were diluted in Laemmli sample buffer [final (1 x) concentration: 31.5 mM 2-Amino-2-(hydroxymethyl)propane-1,3-diol (Tris), 1% (w/v) sodium dodecyl sulfate (SDS), 10% (v/v) glycerol, 0.001% (w/v) 3,3-Bis(3,5-dibromo-4-hydroxyphenyl)–2,1 $\lambda^6$-benzoxathiole-1,1 (3*H*)-dione (bromophenol blue), 5% (v/v) 2-mercaptoethanol; pH 6.8], and denatured at 55 °C for 20 min (unless indicated otherwise). Samples were resolved by SDS-polyacrylamide gel electrophoresis (SDS-PAGE) with self-made Bis(2-hydroxyethyl)amino-tris(hydroxymethyl)methane (BisTris; 250 mM) based 4–20% polyacrylamide (Rotiphorese Gel 30; Carl Roth) gradient gels and run (80–120 V; 90 min) in NuPAGE MOPS SDS running buffer (Thermo Fisher Scientific; Cat#NP000102) using Mini-PROTEAN Tetra Vertical Electrophoresis Cells (Bio-Rad, Cat#1658004). For membrane fractionation experiments, RFP-fluorescence of mDia1 variants in respective fractions was imaged in-gel on a ChemiDoc XRS+ (Bio-Rad) controlled by the Image Lab software (version 6.0.1) utilizing the Alexa546 preset

(605/50 Filter3; Light Green Epi illumination). Separated proteins were wet-blotted (110 V; 90 min; 4 °C) on fluorescence-optimized polyvinylidene difluoride (PVDF) membranes (Immobilon-FL; Merck; IPFL00010) in transfer buffer (25 mM Tris (pH 7.6), 192 mM glycine, 20% (v/v) methanol, 0.03% (w/v) SDS). Subsequently, membranes were blocked with blocking buffer [5% bovine serum albumin (BSA), in Tris-buffered saline (TBS) containing 0.01% Tween 20 (TBS-T)] for 1 hr at RT and incubated (4°C; overnight) with primary antibodies under constant agitation at indicated dilutions (s. Key Resource Table) in blocking buffer. Membranes were washed thrice with TBS-T and incubated with corresponding pairs of IRDye 680RD- or 800CW-conjugated secondary antibodies in TBS-T for 1 hr at RT. After three washes with TBS-T, bound antibodies were visualized by the Odyssey Fc Imaging System (LI-COR Biosciences) controlled and analyzed by Image Studio Lite (Version 5.2.5). For colorimetric analysis of protein levels, the intensity of bands was measured by assigning shapes of equal size to all lanes at similar heights and subtracting the individual background for each shape. Signals were normalized to controls on the same blot. The PageRuler Prestained (Thermo Fisher Scientific; Cat#26616) was used as a ladder to control for protein size.

## Immunoprecipitation from synaptosomes

To immunoprecipitate synaptic mDia1, synaptosomes (P2') were prepared as follows: One mouse brain (p28) was homogenized (900 rpm; 12 strokes with glass-teflon homogenizer) in 7 mL of ice-cold homogenization buffer [(320 mM sucrose, 4 mM HEPES; pH 7.4) supplemented with mammalian protease inhibitor cocktail (PIC)] at 4 °C. Large cellular debris and nuclei were sedimented by centrifugation at 900 g for 10 min at 4 °C. The supernatant was further centrifuged at 12,500 g for 15 min at 4 °C. The resulting pellet (P2) was resuspended in 15 mL of homogenization buffer and pelleted at 12,500 g for 15 min at 4 °C yielding the crude synaptosomal fraction P2'. Subsequently, the pellet was resuspended in 2 mL of immunoprecipitation buffer [20 mM HEPES, 130 mM NaCl, 2 mM MgCl$_2$, 1% (w/v) 3-[Dimethyl[3-(3α,7α,12α-trihydroxy-5β-cholan-24-amido)propyl]azaniumyl]propane-1-sulfonate (CHAPS), PIC, phosphatase inhibitor cocktail II and III (Sigma-Aldrich); pH 7.4] and lysed for 30 min at 4 °C under light agitation. The lysate was cleared by centrifugation at 15,000 g at 4 °C and protein concentration was measured using the BCA assay.

For identification of the protein environment of mDia1, P2' lysate (2 mg; 2 g/L) was incubated with either 2 μg of anti-mDia1 antibody (BD Biosciences, Cat# 610848) or equal amounts of immunoglobulin G (IgG) isotype control mouse antibody (Thermo Fisher Scientific; Cat# 31903; RRID:AB_10959891) for 1 hr at 4 °C before the addition of 25 μL of Pierce Protein A/G Magnetic Beads (Thermo Fisher Scientific) for an additional 2 h under constant rotation. Subsequently, unbound proteins were removed from the beads by three consecutive wash cycles with an immunoprecipitation buffer. Bound proteins were eluted and boiled (10 min; 95 °C) in 2 x Laemmli buffer, resolved by SDS-PAGE, and analyzed by immunoblotting.

## LC-MS analysis of the protein environment of mDia1

For the analysis of interaction partners of synaptic mDia1, mDia1 was immunoprecipitated from synaptosomes as described above. Eluted proteins were reduced (5 mM dithiothreitol; 30 min at 55 °C), alkylated (15 mM iodoacetamide; 20 min at RT in the dark), and submitted to LC-MS analysis: Proteins were subjected to SDS-PAGE following excision of three bands per lane and in-gel digestion of proteins by trypsin (1:100 (w/w); overnight at 37 °C). Resulting tryptic peptides were separated by reverse-phase high-performance liquid chromatography (RP-HPLC; Ultimate 3000 RSLCnano system; Thermo Scientific) using a 50 cm analytical column (in-house packed with Poroshell 120 EC-C18; 2.7 μm; Agilent Technologies) with a 120 min gradient. RP-HPLC was coupled on-line to an Orbitrap Elite mass spectrometer (Thermo Fisher Scientific) that performed precursor ion (MS1) scans at a mass resolution of 60,000, while fragment ion (MS2) scans were acquired with an automatic gain control (AGC) target of $5 \times 10^3$ and a maximum injection time of 50 ms. Data analysis including label-free quantification was performed with MaxQuant (version 1.6.1.0) using the following parameters: The initial maximum mass deviation of the precursor ions was set at 4.5 parts per million (ppm), and the maximum mass deviation of the fragment ions was set at 0.5 Da. Cysteine carbamidometyl and propionamide, Methionine oxidation, and N-terminal acetylation were set as variable modifications. For the identification of proteins, data were searched against the SwissProt murine database (Mouse_2016oktuniprot-proteome%3AUP000000589.fasta). False

discovery rates were <1% at the protein level based on matches to reversed sequences in the concatenated target-decoy database. The statistical analysis was performed utilizing Perseus software (version 1.6.7.0).

## Effector pulldown assays

To analyze the activity of small Rho GTPases, pulldowns utilizing effector protein domains which exclusively bind to their GTP-bound forms were performed: Neurons (200,000 cells) were washed with ice-cold PBS and harvested in GTPase lysis buffer (50 mM HEPES, 500 mM NaCl, 10 mM $MgCl_2$, PIC, phosphatase Inhibitor cocktails II + III; pH 7.4). Cells were lysed for 5 min under repetitive mixing before centrifugation (15,000 g; 5 min; 4 °C). Cleared lysates were incubated with Rhotekin-Rho binding domain (RBD) beads (60 µg; Cytoskeleton Inc) or with PAK-p21 binding domain (PBD) beads (20 µg; Cytoskeleton Inc) at 4 °C under constant rotation to bind active RhoA or Cdc42 and Rac1, respectively. After 2 hr, the beads were pelleted by centrifugation (1000 g; 1 min; 4 °C) and washed with washing buffer (50 mM HEPES, 150 mM NaCl, 10 mM $MgCl_2$; pH 7.4). After repeated centrifugation, unbound proteins were discarded with the supernatant, while bound proteins were eluted from the beads by the addition of 2x Laemmli buffer and boiling (10 min; 95 °C). Finally, the activity of small Rho GTPases was resolved by SDS-PAGE and analyzed by immunoblotting input and pulldown samples.

## Membrane fractionation

To characterize membrane association of the truncation mDia1 mutant, HEK293T cells were transfected with wild-type (mDia1-WT) or mDia1 truncation mutant (mDia1-ΔN) using $CaCl_2$. 48 hr after transfection cells were washed with ice-cold PBS and harvested in ice-cold resuspension buffer (20 mM HEPES, 130 mM NaCl, PIC; pH 7.4). Cells were lysed by forcing the suspension through 18-gauge syringes to crack the plasma membrane in between three freeze-thaw cycles in liquid nitrogen. Nuclei were removed by centrifugation at 1000 g for 5 min at 4 °C. The supernatant (total lysate) was sedimented at 100,000 g for 30 min at 4 °C to yield the membrane fraction as the pellet and the cytosolic fraction as the high-speed supernatant. The pellet was resuspended in a resuspension buffer to the same volume of the cytosolic fraction. Equal volumes of all fractions were analyzed by SDS-PAGE and immunoblotting to allow interpretation of protein enrichment in cytosolic or membrane fractions with respect to the total lysate.

## Statistical analysis

All data in this study are presented as the mean ± standard error of the mean (SEM) and were obtained from $N$ independent experiments with a total sample number of $n$ (e.g. number of images, videos, synapses, etc.) as annotated in the Figure legends. For analysis of protein levels in STED microscopy and synaptic structures in EM, statistical differences between groups were calculated considering $n$, while in all other experiments, statistical differences were calculated between independent experiments $N$ (In pHluorin/CypHer assays, at least 20 responding boutons/video were analyzed). For n>100 or N>5, data were tested for Gaussian distribution following D'Agostino-Pearson tests to determine parametric versus non-parametric statistical testing. The statistical significance between the two groups was evaluated with either two-tailed unpaired student's t-tests for normally distributed data or two-tailed unpaired Mann-Whitney tests, if data did not follow Gaussian distribution. In experiments that necessitated normalization (to 100 or 1) before analysis, one-sample t-tests or one-sample Wilcoxon rank tests were performed for normal and non-normal distributed data, respectively. The statistical significance between more than two experimental groups of normally distributed data was analyzed by one-way ANOVA, followed by a Tukey's post hoc test, while Kruskal-Wallis tests with post hoc Dunn's multiple comparison test were used when datasets did not follow Gaussian distribution.

Corresponding statistical tests are indicated in the Figure and significance levels are annotated as asterisks (*p<0.05, **p<0.01, ***p<0.001, and ****p<0.0001). Differences that are not significant are not stated or indicated as ns (p>0.05). Statistical data evaluation was performed using GraphPad Prism 9.5.1 (733) and all calculated p-values are annotated in corresponding numerical source data files. All Figures were assembled using Affinity Designer (version 1.10.6.1665).

## Contact for reagent and resource sharing

Further information and requests for resources and reagents should be directed to and will be fulfilled by the corresponding contact V.H. (Haucke@fmp-berlin.de).

## Acknowledgements

The authors wish to thank Dr. Art Alberts for sharing the mDia1 KO mice. We are indebted to Sabine Hahn, Delia Löwe, and Silke Zillmann for expert technical assistance with the preparation of neuronal cultures. We further wish to thank Dr. Martin Lehmann (FMP, Berlin), Hannah Gelhaus, and Gresy Bregu (both FU Berlin) for aid with STED imaging, Dr. Dmytro Puchkov (FMP Imaging Core Facility) for supervising electron microscopy analysis, and Heike Stephanowitz and Prof. Fan Liu (FMP Core Facility Proteomics) for proteomic analyses. Supported by grants from the Deutsche Forschungsgemeinschaft (SFB958/TP A01) to VH and NIH/NIA RF1AG050658 and a TIGER grant from the Taub Institute for Research on Alzheimer's Disease and the Aging Brain to FB.

## Additional information

### Funding

| Funder | Grant reference number | Author |
| --- | --- | --- |
| Deutsche Forschungsgemeinschaft | SFB 958/ A01 | Volker Haucke |
| National Institute on Aging | RF1AG050658 | Francesca Bartolini |
| Taub Institute for Research on Alzheimer's Disease and the Aging Brain | TIGER grant | Francesca Bartolini |

The funders had no role in study design, data collection and interpretation, or the decision to submit the work for publication.

### Author contributions

Kristine Oevel, Conceptualization, Investigation, Visualization, Methodology, Writing – original draft; Svea Hohensee, Investigation, Methodology; Atul Kumar, Resources, Methodology; Irving Rosas-Brugada, Investigation, Visualization; Francesca Bartolini, Resources, Supervision, Methodology; Tolga Soykan, Conceptualization, Supervision, Investigation, Visualization, Methodology, Writing – original draft, Writing – review and editing; Volker Haucke, Conceptualization, Resources, Supervision, Funding acquisition, Validation, Visualization, Writing – original draft, Project administration, Writing – review and editing

### Author ORCIDs

Kristine Oevel ⓘ http://orcid.org/0009-0003-3671-5411
Tolga Soykan ⓘ http://orcid.org/0000-0002-1324-9601
Volker Haucke ⓘ https://orcid.org/0000-0003-3119-6993

### Ethics

All animal experiments were reviewed and approved by the ethics committee of the Landesamt für Gesundheit und Soziales (LAGeSo) Berlin or the Committee on the Ethics of Animal Experiments of Columbia University and conducted according to the committees' guidelines (LAGeSo) or the Guide for the Care and Use of Laboratory Animals of the National Institutes of Health (for mDia1 KO mice).

Reviewer 1 Public Review: https://doi.org/10.7554/eLife.92755.3.sa1
Reviewer 2 Public Review: https://doi.org/10.7554/eLife.92755.3.sa2
Author Response https://doi.org/10.7554/eLife.92755.3.sa3

## Additional files

### Supplementary files
- MDAR checklist
- Supplementary file 1. List of oligonucleotides used in this study.

### Data availability
All data generated or analysed during this study are included in the manuscript and supporting files; source data files have been provided.

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

# Appendix 1

**Appendix 1—key resources table**

| Reagent type (species) or resource | Designation | Source or reference | Identifiers | Additional information |
|---|---|---|---|---|
| Genetic reagent (*Mus musculus*) | C57BL/6 N wild-type | Leibniz Research Institute for Molecular Pharmacology | RRID:IMSR_JAX:000664 | |
| Genetic reagent (*Mus musculus*) | mDia1 KO | Bartolini Lab, Columbia University | *Peng et al., 2003*; https://doi.org/10.1016/s0960-9822(03)00170–2 | Genetic knockout of Diaph1 |
| Cell line (*Homo sapiens*) | HEK293T | American Type Culture Collection | Cat# CRL-3216; RRID:CVCL_0063 | |
| Antibody | β-Actin (mouse monoclonal) | Sigma-Aldrich | Cat# A5441; RRID:AB_476744 | IB (1:5000) |
| Antibody | Bassoon (guinea pig monoclonal) | Synaptic Systems | Cat# 141 318; RRID:AB_2927388 | IC (1:100) |
| Antibody | CDC42 (rabbit polyclonal) | Abcam | Cat# ab64533; RRID:AB_1310067 | IB (1:1000); IC (1:100) |
| Antibody | Dynamin1 (rabbit polyclonal) | Pietro D. Camilli | *Shupliakov et al., 2002*; DOI: 10.1126/science.276.5310.29 | IB (1:2000) |
| Antibody | GFP (mouse monoclonal) | Thermo Fisher Scientific | Cat# A-11120; RRID:AB_221568 | IC (1:2500) |
| Antibody | Homer1 (mouse monoclonal) | Synaptic Systems | Cat# 160 011; RRID:AB_2120992 | IC (1:200) |
| Antibody | Homer1 (rabbit polyclonal) | Synaptic Systems | Cat# 160 003; RRID:AB_887730 | IC (1:200) |
| Antibody | LAMP1 (rabbit monoclonal) | Cell Signaling Technology | Cat# 9091; RRID:AB_2687579 | IB (1:1000) |
| Antibody | mDia1 (mouse monoclonal) | BD Biosciences | Cat# 610848; RRID:AB_398167 | IB (1:500); IP (2 µg) |
| Antibody | mDia1 (rabbit monoclonal) | Abcam | Cat# ab129167; RRID:AB_11143749 | IC (1:100) |
| Antibody | mDia3 (rabbit polyclonal) | Sigma-Aldrich | Cat# HPA005647; RRID:AB_1078657 | IB (1:1000) |
| Antibody | Myosin IIb (mouse monoclonal) | Abcam | Cat# ab684; RRID:AB_305661 | IC (1:100) |
| Antibody | Myosin IIb (rabbit polyclonal) | Cell Signaling Technology | Cat# 3404; RRID:AB_126421 | IB (1:2000) |
| Antibody | Rac1 (mouse monoclonal) | BD Biosciences | Cat# 610650; RRID:AB_397977 | IB (1:1000); IC (1:50) |
| Antibody | RFP (rabbit polyclonal) | Takara Bio | Cat# 632496; RRID:AB_10013483 | IC (1:500) |
| Antibody | RhoA (rabbit monoclonal) | Cell Signaling Technology | Cat# 2117; RRID:AB_10693922 | IB (1:1000); IC (1:100) |
| Antibody | α-Tubulin (mouse monoclonal) | Sigma-Aldrich | Cat# T9026; RRID:AB_47759 | IB (1:5000) |
| Antibody | IgG control (mouse monoclonal) | Thermo Fisher Scientific | Cat# 31903; RRID:AB_10959891 | IP (2 µg) |
| Antibody | vGAT-CypHer5E (rabbit polyclonal) | Synaptic Systems | Cat# 131 103CpH; RRID:AB_2189809 | Live-imaging (1:500) |
| Antibody | Anti-mouse IgG Atto542 (donkey polyclonal) | Martin Lehmann | *Gonschior et al., 2022*; https://doi.org/10.1038/s41467-022-32533-4 | IC (1:400) |
| Antibody | Anti-rabbit IgG Atto542 (donkey polyclonal) | Martin Lehmann | *Gonschior et al., 2022*; https://doi.org/10.1038/s41467-022-32533-4 | IC (1:400) |
| Antibody | Anti-mouse Alexa Fluor 594 (goat polyclonal) | Thermo Fisher Scientific | Cat# A-11032; RRID:AB_2534091 | IC (1:200) |
| Antibody | Anti-rabbit Alexa Fluor 594 (goat polyclonal) | Thermo Fisher Scientific | Cat# A-11037; RRID:AB_2534095 | IC (1:200) |
| Antibody | Anti-guinea pig Atto647N (camelid monoclonal) | Synaptic Systems | Cat# N0602-At647N-S; RRID:AB_2744576 | IC (1:200) |
| Antibody | Anti-mouse Alexa Fluor 647 (goat polyclonal) | Thermo Fisher Scientific | Cat# A-21235; RRID:AB_2535804 | IC (1:200) |
| Antibody | Anti-mouse IRDye 680RD (goat polyclonal) | LI-COR Biosciences | Cat# 925–68070; RRID:AB_2651128 | IB (1:10000) |

*Appendix 1 Continued on next page*

*Appendix 1 Continued*

| Reagent type (species) or resource | Designation | Source or reference | Identifiers | Additional information |
|---|---|---|---|---|
| Antibody | Anti-rabbit IRDye 680RD (goat polyclonal) | LI-COR Biosciences | Cat# 926–68071; RRID:AB_10956166 | IB (1:10000) |
| Antibody | Anti-mouse IgG, IRDye 800CW (goat polyclonal) | LI-COR Biosciences | Cat# 926–32210; RRID:AB_621842 | IB (1:10000) |
| Antibody | Anti-rabbit IgG, IRDye 800CW (goat polyclonal) | LI-COR Biosciences | Cat# 926–32211; RRID:AB_621843 | IB (1:10000) |
| Chemical compound, drug | Phalloidin-Alexa Fluor 594 | AAT Bioquest | Cat# ABD-23158 | IC (1:1000) |
| Recombinant DNA reagent | Synaptophysin-pHluorin | Leon Lagnado | *Gonschior et al., 2022*; https://doi.org/10.1038/s41467-022-32533-4 | Expresses rat Synaptophysin-2xpHluorin (inserted between $Asn_{183}$ – $Thr_{184}$) under a CMV promotor |
| Recombinant DNA reagent | vGlut1-pHluorin | Volker Haucke | *Bolz et al., 2023*; https://doi.org/10.1016/j.neuron.2023.08.016 | Expresses rat vGlut1-pHluorin (inserted between $Gly_{99}$ – $Gly_{100}$) under a hSyn1 promotor; lentiviral plasmid |
| Recombinant DNA reagent | mCherry | Clontech | Cat# 632523 | Expresses mCherry under a CMV promotor |
| Recombinant DNA reagent | mDia1-WT-mCherry | This study | - | Expresses mouse mDia1-WT-mCherry under a CMV promotor |
| Recombinant DNA reagent | mDia1-ΔN -mCherry | This study | - | Expresses truncation (first 60 AA) variant of mouse mDia1-mCherry under a CMV promotor |
| Recombinant DNA reagent | mDia1-WT-SNAP | This study | - | Expresses mouse mDia1-WT-SNAP under a hSyn1 promotor; lentiviral plasmid |
| Recombinant DNA reagent | mDia1-K994A-SNAP | This study | - | Expresses K994A variant of mDia1-SNAP under a hSyn1 promotor; lentiviral plasmid |
| Recombinant DNA reagent | Dynamin1-WT-SNAP | This study | - | Expresses mouse Dynamin1-WT-SNAP under a hSyn1 promotor; lentiviral plasmid |
| Recombinant DNA reagent | Dynamin1-K44A-SNAP | This study | - | Expresses K44A variant of mouse Dynamin1-SNAP under a hSyn1 promotor; lentiviral plasmid |

*Appendix 1 Continued on next page*

*Appendix 1 Continued*

| Reagent type (species) or resource | Designation | Source or reference | Identifiers | Additional information |
|---|---|---|---|---|
| Recombinant DNA reagent | vGlut1-mCherry | Franck Polleux | *Kwon et al., 2016*; https://doi.org/10.1371/journal.pbio.1002516 | Expresses rat vGlut1-mCherry under a CAG promotor |
| Recombinant DNA reagent | Rac1-CA | Addgene | Cat# 12983; RRID:Addgene_12983 | Expresses Q61L variant of human myc-Rac1 under a CMV promotor |
| Recombinant DNA reagent | Rac1-DN | Addgene | Cat# 12984; RRID:Addgene_12984 | Expresses T17N variant of human myc-Rac1 under a CMV promotor |
| Recombinant DNA reagent | RhoA-WT | Theofilos Papadopoulos | *Reddy-Alla et al., 2010*; https://doi.org/10.1111/j.1460-9568.2010.07149.x | Expresses human 3xHA-RhoA-WT under a CMV promotor |
| Recombinant DNA reagent | RhoA-DN | Theofilos Papadopoulos | *Reddy-Alla et al., 2010*; https://doi.org/10.1111/j.1460-9568.2010.07149.x | Expresses T19N variant of human 3xHA-RhoA under a CMV promotor |
| Recombinant DNA reagent | RhoB-WT | Harry Mellor | *Mellor, 1998*; https://doi.org/10.1074/jbc.273.9.4811 | Expresses human myc-RhoB-WT under a CMV promotor |
| Recombinant DNA reagent | RhoB-DN | Harry Mellor | *Mellor, 1998*; https://doi.org/10.1074/jbc.273.9.4811 | Expresses T19N variant of human myc-RhoB under a CMV promotor |
| Recombinant DNA reagent | shCTR (transfected) | Sigma-Aldrich | Cat# SHC016 | No murine targets |
| Recombinant DNA reagent | shmDia1 (transfected) | Sigma-Aldrich | Cat# TRCN0000108685 | Targets 3'UTR of NM_007858 |
| Recombinant DNA reagent | shRhoA (transfected) | Sigma-Aldrich | Cat# TRCN0000302388 | Targets CDS of NM_016802 |
| Recombinant DNA reagent | shRhoB (transfected) | Sigma-Aldrich | Cat# TRCN0000294874 | Targets CDS of NM_007483 |
| Recombinant DNA reagent | shCTRmiR | This study | - | Expresses rat Synaptophysin-2xpHluorin under a CMV promotor and shRNA embedded into a microRNA (shRNAmiR) with no murine targets |
| Recombinant DNA reagent | shmDia1miR | This study | - | Expresses rat Synaptophysin-2xpHluorin under a CMV promotor and shRNAmiR against the CDS of mouse Diaph1 (mDia1) |

*Appendix 1 Continued on next page*

*Appendix 1 Continued*

| Reagent type (species) or resource | Designation | Source or reference | Identifiers | Additional information |
|---|---|---|---|---|
| Recombinant DNA reagent | shCTR (transduced) | Christian Rosenmund | *Watanabe et al., 2014*; https://doi.org/10.1038/nature13846 | Expresses NLS-RFP or BFP under a hSyn1 promotor and shRNA against no murine target (msClathrin scrambled) under a U6 promotor |
| Recombinant DNA reagent | shmDia1 (transduced) | This study | - | Expresses NLS-RFP or BFP under a hSyn1 promotor and shRNA against the 3'UTR of mouse Diaph1 (mDia1) under a U6 promotor |
| Recombinant DNA reagent | shmDia3 (transduced) | This study | - | Expresses NLS-RFP or BFP under a hSyn1 promotor and shRNA against the CDS of mouse Diaph2 (mDia3) under a U6 promotor |
| Recombinant DNA reagent | MD2.G | Addgene | Cat# 12259; RRID:Addgene_12259 | Expresses lentiviral VSV-G envelope protein |
| Recombinant DNA reagent | psPAX2 | Addgene | Cat# 12260; RRID:Addgene_12260 | Expresses lentiviral packaging protein |
| Recombinant DNA reagent | pOrange GFP-β-Actin KI | Addgene | Cat#131479; RRID:Addgene_131479 | gRNA and GFP donor for endogenous N-terminal tagging of β-Actin (amino acid position: D2) by targeting Actb |
| Chemical compound, drug | Dynasore | Sigma-Aldrich | Cat# D7693 | 80 µM 10 min |
| Chemical compound, drug | EHT 1864 | MedChemExpress | Cat# HY-16659 | 10 µM acute (CypHer); 2 h (IC/EM) |
| Chemical compound, drug | IMM-01 | Sigma-Aldrich | Cat# SML1064 | 10 µM acute |
| Chemical compound, drug | Jasplakinolide | Sigma-Aldrich | Cat# J4580 | 8 µM acute (JLY); 1 µM 30 min |
| Chemical compound, drug | Latrunculin A | Cayman Chemical | Cat# CAY10684 | 5 µM acute (JLY) |
| Chemical compound, drug | ML141 | MedChemExpress | Cat# HY-12755 | 10 µM acute |
| Chemical compound, drug | Rhosin | MedChemExpress | Cat# HY-12646 | 10 µM 2 h |
| Chemical compound, drug | Tetrodotoxin | Carl Roth | Cat# 6973.1 | 1 µM 36 h |
| Chemical compound, drug | Y-27632 | Tocris | Cat# 1254 | 1 µM acute (JLY) |

*Appendix 1 Continued on next page*

*Appendix 1 Continued*

| Reagent type (species) or resource | Designation | Source or reference | Identifiers | Additional information |
|---|---|---|---|---|
| Commercial assay or kit | Rhotekin-Rho binding domain (RBD) beads | Cytoskeleton Inc | Cat# RT02 | 60 µg |
| Commercial assay or kit | PAK-p21 binding domain (PBD) beads | Cytoskeleton Inc | Cat# PAK02 | 20 µg |
| Commercial assay or kit | ProFection Mammalian Transfection System – Calcium Phosphate | Promega | Cat# E1200 | |
| Commercial assay or kit | Gibson Master Mix | New England Biolabs Inc | Cat# E2611L | |
| Commercial assay or kit | Q5 site-directed mutagenesis kit | New England Biolabs Inc | Cat# E0552S | |
| Software, algorithm | Fiji | NIH | RRID:SCR_002285 | |
| Software, algorithm | Prism | GraphPad | RRID:SCR_002798 | Version 9.5.1. |
| Software, algorithm | Image Lab Software | Bio-Rad | RRID:SCR_014210 | Version 6.0.1 |
| Software, algorithm | Image Studio Lite | LI-COR Biosciences | RRID:SCR_013715 | Version 5.2.5 |
| Software, algorithm | pHluorin ROI sector | Github | https://github.com/DennisVoll/pHluorin_ROI_selector/; *Voll, 2020* | |
| Software, algorithm | SynActJ | Martin Lehmann | *Schmied et al., 2021*; https://doi.org/10.3389/fcomp.2021.777837 | |
| Software, algorithm | Macro_plot_lineprofile_multicolor | Kees Straatman | *Gerth et al., 2017*; https://doi.org/10.1016/j.str.2019.03.020 | |
| Software, algorithm | MaxQuant | Jürgen Cox | https://www.maxquant.org/maxquant/ | Version 1.6.1.0 |
| Software, algorithm | Perseus | Jürgen Cox | https://www.maxquant.org/perseus/ | Version 1.6.7.0 |
| Sequence-based reagent | see *Supplementary file 1* | | | |

