## [Editor Report · eLife assessment]

This manuscript provides **convincing** evidence for the involvement of membrane actin, and its regulatory proteins, mDia1/3, RhoA, and Rac1 in the mechanism of synaptic vesicle re-uptake (endocytosis). These **important** data fill a gap in the understanding of how the regulation of actin dynamics and endocytosis are linked. The manuscript will be of interest to all scientists working on cellular trafficking and membrane remodeling.

---

## [Referee Report · Reviewer 1 Public Review]

Summary:

The authors set out to clarify the molecular mechanism of endocytosis (re-uptake) of synaptic vesicle (SV) membrane in the presynaptic terminal following release. They have examined the role of presynaptic actin, and of the actin regulatory proteins diaphanous-related formins ( mDia1/3), and Rho and Rac GTPases in controlling the endocytosis. They successfully show that presynaptic membrane-associated actin is required for normal SV endocytosis in the presynaptic terminal, and that the rate of endocytosis is increased by activation of mDia1/3. They show that RhoA activity and Rac1 activity act in a partially redundant and synergistic fashion together with mDia1/3 to regulate the rate of SV endocytosis. The work adds substantially to our understanding of the molecular mechanisms of SV endocytosis in the presynaptic terminal.

Strengths:

The authors use state-of-the-art optical recording of presynaptic endocytosis in primary hippocampal neurons, combined with well-executed genetic and pharmacological perturbations to document effects of alteration of actin polymerization on the rate of SV endocytosis. They show that removal of the short amino-terminal portion of mDia1 that associates with the membrane interrupts the association of mDia1 with membrane actin in the presynaptic terminal. They then use a wide variety of controlled perturbations, including genetic modification of the amount of mDia1/3 by knock-down and knockout, combined with inhibition of activity of RhoA and Rac1 by pharmacological agents, to document the quantitative importance of each agent, and their synergistic relationship in regulation of endocytosis.

The analysis is augmented by ultrastructural analyses that demonstrate the quantitative changes in numbers of synaptic vesicles and in uncoated membrane invaginations that are predicted by the optical recordings.

The manuscript is well-written and the data are clearly explained. Statistical analysis of the data is strengthened by the very large number of data points analyzed for each experiment.

Weaknesses:

There are no major weaknesses.

---

## [Referee Report · Reviewer 2 Public Review]

Summary:

This manuscript expands previous work from the Haucke group which demonstrated the role of formins in synaptic vesicle endocytosis. The techniques used to address the research question are state-of-the-art. As stated above there is a significant advance in knowledge, with particular respect to Rho/Rac signalling.

Strengths:

The major strength of the work was to reveal new information regarding the control of both presynaptic actin dynamics and synaptic vesicle endocytosis via Rho/Rac cascades. In addition, there was further mechanistic insight regarding the specific function of mDia1/3. The methods used were state-of-the-art.

Weaknesses:

There are no major weaknesses.

---

## [Author Response]

The following is the authors’ response to the original reviews.

**Public reviews**

**Reviewer 1 (Public Review):**
Summary:The authors set out to clarify the molecular mechanism of endocytosis (re-uptake) of synaptic vesicle (SV) membrane in the presynaptic terminal following release. They have examined the role of presynaptic actin, and of the actin regulatory proteins diaphanous-related formins (mDia1/3), and Rho and Rac GTPases in controlling the endocytosis. They successfully show that presynaptic membrane-associated actin is required for normal SV endocytosis in the presynaptic terminal and that the rate of endocytosis is increased by activation of mDia1/3. They show that RhoA activity and Rac1 activity act in a partially redundant and synergistic fashion together with mDia1/3 to regulate the rate of SV endocytosis. The work adds substantially to our understanding of the molecular mechanisms of SV endocytosis in the presynaptic terminal.Strengths:The authors use state-of-the-art optical recording of presynaptic endocytosis in primary hippocampal neurons, combined with well-executed genetic and pharmacological perturbations to document effects of alteration of actin polymerization on the rate of SV endocytosis. They show that removal of the short amino-terminal portion of mDia1 that associates with the membrane interrupts the association of mDia1 with membrane actin in the presynaptic terminal. They then use a wide variety of controlled perturbations, including genetic modification of the amount of mDia1/3 by knock-down and knockout, combined with inhibition of activity of RhoA and Rac1 by pharmacological agents, to document the quantitative importance of each agent and their synergistic relationship in regulation of endocytosis.The analysis is augmented by ultrastructural analyses that demonstrate the quantitative changes in numbers of synaptic vesicles and in uncoated membrane invaginations that are predicted by the optical recordings.The manuscript is well-written and the data are clearly explained. Statistical analysis of the data is strengthened by the very large number of data points analyzed for each experiment.Weaknesses:There are no major weaknesses. The optical images as first presented are small and it is recommended that the authors provide larger, higher-resolution images.

Response: We thank the referee for these highly positive remarks. In response, we now provide larger, high-resolution images as requested.

**Reviewer 2 (Public Review):**
Summary:This manuscript expands on previous work from the Haucke group which demonstrated the role of formins in synaptic vesicle endocytosis. The techniques used to address the research question are state-of-the-art. As stated above there is a significant advance in knowledge, with particular respect to Rho/Rac signalling.Strengths:The major strength of the work was to reveal new information regarding the control of both presynaptic actin dynamics and synaptic vesicle endocytosis via Rho/Rac cascades. In addition, there was further mechanistic insight regarding the specific function of mDia1/3. The methods used were state-of-the-art.Weaknesses:There are a number of instances where the conclusions drawn are not supported by the submitted data, or further work is required to confirm these conclusions.

Response: We thank the referee for his/her thorough reading of the manuscript and the thoughtful comments and questions. We have conducted additional experiments and made textual change to our manuscript to address these points and to further strengthen the conclusions as detailed in our response to the recommendations for authors.

**Recommendations for the authors**

**Reviewer 1 (Recommendations For The Authors):**
Most of the figures contain images that are too small to be easily interpreted because the resolution is degraded when they are enlarged in the PDF file. The authors should redesign the figures so that the letters marking each panel are smaller, and the size of each data panel is much larger (at least twice as large with increased resolution). There is, at present, a great deal of white space in most of the figures that should be reduced to make room for larger, higher-resolution images. Larger fonts should be used for annotations of the images so that they are easier to read. The data appears to be very high quality, but it is presented at a size and resolution that don't do it justice.

Response: We thank the referee for his/ her helpful comments. In response to the referee’s comment, we have carefully re-arranged all figures and now provide larger, high-resolution images.

**Reviewer 2 (Recommendations For The Authors):**
Major points(1) Figure 1 - While there is a rationale for employing a cocktail of drugs to interfere with actin dynamics, it would be highly informative to determine the effect of these modulators in isolation. This is important, since in their previous publication (Soykan et al Neuron 2017 93:854) the authors demonstrated that latrunculin had no effect, while jasplakinolide accelerated endocytosis of originating purely from Y-27362 and ROCK kinase inhibition, rather than destabilisation/stabilisation of actin. It will be key to dissect this by examining the effect on endocytosis of both (1) a cocktail of latrunculin/jasplakinolide and (2) Y-27362 alone.

Response: We thank the referee for highlighting this interesting point. We have now experimentally addressed the effect of latrunculin (L), jasplakinolide (J) and the ROCK inhibitor Y-27362 (Y) either alone or in combination on the kinetics of synaptic vesicle (SV) endocytosis(new Fig. 1-Supplement 1C,D). We now demonstrate that application of the ROCK inhibitor Y-27362 or the combination of latrunculin (L) and jasplakinolide (J) have no effect on Syph-pH endocytosis. Combined use of jasplakinolide (J) and the ROCK inhibitor Y-27362 (Y) has a small phenotype. In contrast, a mix of all three inhibitors (JYL) potently impairs endocytosis kinetics at hippocampal synapses. These data demonstrate that actin dynamics are required for SV endocytosis, while ROCK inhibition alone does not appear to impair endocytosis kinetics. We note that our data are in line with a study by Ann Saal et al (2020) who reported a lack of effect of ROCK inhibition on the kinetics of Synaptotagmin1-CypHer retrieval.

(2) Figure 1 - There are clear effects on the retrieval of pHluorin reporters and also endogenous vGAT in the presence of disruptors of actin function. However, there was no assessment of the impact of these interventions on either neurotransmitter release or SV fusion (with the exception of 1 condition with one stimulus train (Fig S1D), and the effect of Rac modulation in Fig S6F). As quoted by the authors, previous studies using knockout of beta- or gamma-actin have shown a profound effect on these parameters in hippocampal neurons, which has the potential to impact the speed and extent of compensatory endocytosis. The authors will already have this data from the use of the two reporters (pHluorn and GAT-cypHer), and it is important to include this to allow interpretation of the effect on endocytosis observed.

Response: We agree with the referee that this is an important point that we have tackled experimentally using vGAT-CypHer and synapto-pHluorin responses as measures. In the new Fig. 1-Supplement 1, Fig. 5- Supplement 1, and Fig.6 -Supplement 1 of our revised manuscript, we show that SV exocytosis is largely unaffected by any of the applied manipulations of actin function.

Specifically, we have added surface normalized data as a surrogate measure for exocytosis for the following:

JLY treatment monitored by Syph-pH (Figure 1-Supplement 1A) and vGAT-CypHer (Figure 1-Supplement 1B),shCTR/shmDia1 (transfected) assayed via Syph-pH (Figure 1-Supplement 1G),shCTR/shmDia1/shmDia1+3 assayed via vGLUT1-pH (40AP: Figure 1-Supplement 1J; 80AP: Figure 1-Supplement 1L),shCTR/shmDia1+3 (transduced) assayed by vGAT-CypHer (Figure 1-Supplement 1M),IMM treatment monitored by vGLUT1-pH (Figure 1-Supplement 1O),RhoA/B WT/DN overexpression monitored by Syph-pH (Figure 5-Supplement 1B),shCTR/shRhoA+B (transfected) monitored via Syph-pH (Figure 5-Supplement 1D),shCTR/shmDia1+3 +/- EHT 1864 (Rac Inhibitor) assayed by vGAT-CypHer (Figure 6-Supplement 1D),shCTR/shmDia1+3 +/- Rac1-CA/DN assayed by Syph-pH (Figure 6-Supplement 1F).

The lack of effect of these manipulations on exocytic SV fusion is thus distinct from the effects of complete abrogation of actin expression in beta- or gamma-actin knockout studies reported by the LingGang Wu laboratory (Neuron 2016) as the referee also noted.

(3) Figure 3H, 3K, 4C, 4F - It is unclear how the values on the Y-axis were calculated. Regardless, to confirm that there is a specific increase in presynaptic mDia1/actin, the equivalent values for Homer/mDia1 should be presented (with Basson/Homer as a negative control). Without this, it is difficult to argue for a specific enrichment of mDia1/actin at the presynapse. The CRISPR experiments help with this interpretation (Fig 4G-I), however, inclusion of the Homer/mDia1 STED data would strengthen it greatly.

Response: We apologize if the description has been unclear. We essentially have followed the same type of analysis as recently described by Bolz et al (2023). In brief, the rationale for quantifying presynaptic protein levels of interests is as follows: The presynaptic area was defined by the normalized distribution curve of Bassoon, i.e. area between 151.37 and -37.84 nm as marked by purple shading with a cutoff set where Bassoon and Homer1 distributions overlap (-37.84 nm) as shown in Figure 3Supplement 1H (pasted below). The individual synaptic line profiles, e.g. of mDia1 were integrated to yield presynaptic between 151.37 and -37.84 nm (purple in the graph) vs. postsynaptic levels from - 56.76 to -245.97 nm (green shaded area). new Figure 3-Supplement 1H-J

**Author response image 1. sa3fig1:** Based on this analysis postsynaptic mDia1 levels were also elevated upon Dynasore treatment (new Figure 3-Supplement 1I). In spite of this and consistent with the fact that the majority of mDia1 is localized at the presynapse, we found that postsynaptic F-actin levels were unchanged in mDia1/3depleted neurons (p = 0.0966; One sample t-test) (new Figure 4-Supplement 1E,F).

**Author response image 2. sa3fig2:** Moreover, we also conducted further analysis with respect to possible effects of Dynasore on synaptic architecture in general. Neither presynaptic Bassoon nor postsynaptic Homer1 levels were significantly altered by Dynasore treatment (new Figure 3–Supplement 1J).

(4) Figure 4J - The rescue of the pHlourin response by jasplakinolide is difficult to interpret when considering previous work from the same authors. In their 2017 publication (Soykan et al Neuron 2017 93:854), they revealed that the drug accelerated the pHluorin response, whereas now they demonstrate no effect in the control condition. If the drug does accelerate endocytosis, then it may be working via a different mechanism to restore endocytosis in mDia1/3 knockdown neurons.

Response: The referee is correct. The very mild acceleration of endocytosis in the presence of jasplakinolide can be observed using synaptophysin-pHluorin as a reporter under moderate mediumfrequency stimulation at 10Hz for 5 s (i.e. 50 APs). In the present dataset using a different pHluorin reporter (i.e. vGLUT1-pHluorin) that tends to yield faster endocytic responses (as noted before by the Ryan lab) and using a high frequency stimulus (20Hz) we fail to observe a significant effect. While this cannot be excluded, we would be reluctant to conclude that these differences indicate distinct mechanisms of jasplakinolide action. Alternatively, actin may be of particular importance under conditions of high-frequency stimulation.

In this regard, the conclusions from the pHluorin experiment would be greatly strengthened by demonstrating that jasplakinolide corrects the reduction of presynaptic actin in mDia1/3 knockdown synapses observed in figures 4E-I.

Response: As demonstrated in Figure 4-Supplement 1G and in support of a common mechanism of action, we find that application of jasplakinolide rescues reduced presynaptic actin levels in mDia1/3depleted neurons. The respective data for presynaptic actin (normalized to shCTR + DMSO set to 100) are: shCTR + DMSO = 100 ± 6.3; shmDia1+3 + DMSO = 47.7 ± 4.3; shCTR + Jasp = 150.6 ± 11.9; shmDia1+3 + Jasp = 94.3 ± 11.5.These data are now also quoted in the revised manuscript text.

Minor points(1) There is no rationale provided regarding why different stimulation protocols are sometimes used in the pHluorin/cypHer experiments. In most cases it is 200 APs (40 Hz), however, in some cases, it is 40 APs or 80 APs. Can the authors explain why they used these different protocols?

Response: The referee noted this correctly. This in part reflects the history of the project, in which initial datasets were acquired using 200 AP trains using pHluorin reporters. To probe whether the phenotypic effects induced by actin perturbations, were robust over different stimulation paradigms and optical reporters, additional data using either 40 or 80 AP trains as well as experiments capitalizing on vGLUT1 or endogenous vGAT monitiored by pH-sensitive cypHer-labeled antibodies were conducted. We hope the referee agrees that these additional data add to the general importance of our study.

(2) Figure 2 - The reduction in SV density in mDia1/3 knockdown neurons correlates with the results in Figures 1 and 7. However, a functional consequence of this reduction (change in size of RRP or neurotransmitter release, as stated above) would have increased the impact of these experiments.

Response: We agree with the referee and will address this interesting possibility using electrophysiolgical recordings in future studies.

(3) It appears the experimental n in Figure 2 is profiles, rather than experiments. This should be clarified, especially since there is no reference to how many times the experiments in Fig2E-G were performed.

Response: This point has been clarified in the revised figure legend.

(4) Figure 6 - The authors state that inhibition of Rac function either via a dominant negative mutant or an inhibitor increases the inhibition of endocytosis via knockdown of mDia1/3. However, both interventions inhibit endocytosis themselves in the control condition. It would be informative to see the full statistical analysis of this data since there does not appear to be a significant additive effect when comparing Rac inhibition with the additional knockdown of mDia1/3.

Response: In our revised manuscript, we now provide the full statistical analysis in the revised Source Data Table for Figures 6G,H. We observe that Rac1-DN expression indeed further aggravates phenotypes elicited by depletion of mDia1+3, but not vice versa. We have modified the corresponding section in the results section of our revised manuscript accordingly.

(5) Figure 7 - The increase in endosomes in mDia1/3 knockdown neurons is consistent with previous studies examining pharmacological inhibition of formins (Soykan et al Neuron 2017 93:854). However, it is noted that these structures were absent in the images shown in Figure 2. Similar to the previous point in figure 6, a full reporting of the significance of different conditions is important here, since it appears that the only difference between EHT1864 and its co-incubation with mDia1/3 knockdown neurons is in the number of ELVs (Fig 7H).

Response: Similar to the example EM images shown in Figure 7, enlarged endocytic structures are also observed in shmDia1+3 depleted synapses shown in Figure 2. However, ELVs and membrane invaginations were not color-coded as the focus in figure 2 is on the reduction of the SV pool. To better illustrate this, we have chosen a more representative example of this phenotype in revised Figure 2.

Moreover, we now provide the full statistical analysis of EM phenotypes in the revised Source Data Table for Figure 7. We find that Rac1 inhibition indeed significantly aggravates the effects of mDia1+3 loss with respect to the accumulation of membrane invaginations, while the effect on ELVs remains insignificant. However, accumulation of ELVs in the presence of the Rac1 inhibitor EHT1864 is further aggravated upon depletion of mDia1+3. We have modified the corresponding section in the results section of our revised manuscript accordingly.

We speculate that Rac1 may thus predominantly act at the plasma membrane, whereas mDia1/3 may serve additional functions in SV reformation at the level of ELVs. Clearly, further studies would be needed to test this idea in the future.